# An unsupervised learning approach to identifying blocking events: the case of European summer

Carl Thomas[1], Apostolos Voulgarakis[1,2], Gerald Lim[3], Joanna Haigh[1,4], and Peer Nowack[1,4,5,6]

[1]Department of Physics, Imperial College London, South Kensington Campus, London, SW7 2BW, UK
[2]School of Environmental Engineering, Technical University of Crete, Chania, Crete, 73100, Greece
[3]Centre for Climate Research Singapore, 36 Kim Chuan Road, 537054, Singapore
[4]Grantham Institute, Imperial College London, SW7 2AZ, UK
[5]Climatic Research Unit, School of Environmental Sciences, Norwich, NR4 7TJ, UK
[6]Data Science Institute, Imperial College London, South Kensington Campus, London, SW7 2AZ, UK

**Correspondence:** Carl Thomas (c.thomas18@imperial.ac.uk)

**Abstract.** Atmospheric blocking events are mid-latitude weather patterns, which obstruct the usual path of the polar jet streams. They are often associated with heat waves in summer and cold snaps in winter. Despite being central features of mid-latitude synoptic-scale weather, there is no well-defined historical dataset of blocking events. Various blocking indices (BIs) have thus been suggested for automatically identifying blocking events in observational and in climate model data. However, BIs show significant regional and seasonal differences so that several indices are typically applied in combination to ensure scientific robustness. Here, we introduce a new BI using self-organizing maps (SOMs), an unsupervised machine learning approach, and compare its detection skill to some of the most widely applied BIs. To enable this intercomparison, we first create a new ground truth time series classification of European blocking based on expert judgement. We then demonstrate that our method (SOM-BI) has several key advantages over previous BIs because it exploits all of the spatial information provided in the input data and reduces the dependence on arbitrary thresholds. Using ERA5 reanalysis data (1979-2019), we find that the SOM-BI identifies blocking events with a higher precision and recall than other BIs. In particular, SOM-BI already performs well using only around 20 years of training data so that observational records are long enough to train our new method. We present case studies of the 2003 and 2019 European heat waves and highlight that well-defined groups of SOM nodes can be an effective tool to diagnose such weather events, although the domain-based approach can still lead to errors in the identification of certain events in a fashion similar to the other BIs. We further test the red blocking detection skill of SOM-BI depending on the meteorological variable used to study blocking, including geopotential height, sea level pressure and four variables related to potential vorticity, and the 500 hPa geopotential height anomaly field provides the best results with our new approach. We also demonstrate how SOM-BI can be used to identify different types of blocking events and their associated trends. Finally, we evaluate the SOM-BI performance on around 100 years of climate model data from a pre-industrial simulation with the new UK Earth System Model (UKESM1-0-LL). For the model data, all blocking detection methods have lower skill than for the ERA5 reanalysis, but SOM-BI performs noticeably better than the conventional indices. Overall, our results demonstrate the significant potential for unsupervised learning to complement the study of blocking events in both reanalysis and climate modelling contexts.

## 25    1    Introduction

Atmospheric blocking events are large-scale mid-latitude anticyclones that can persist for several days, which obstruct the typical westerly flow pattern (Rex, 1950). Blocking systems are often associated with regional extreme weather events, particularly heat waves in summer and cold snaps in winter. For example, the 2003 summer heat wave and 2009/10 winter cold events in Europe were both associated with atmospheric blocking (Black et al., 2004; Cattiaux et al., 2010). The evolution of
atmospheric blocking itself is nonlinear (Palmer, 1999) and the underlying complex physical mechanisms are not yet understood (Nakamura and Huang, 2018; Woollings et al., 2018). There is a large seasonal, inter-annual and decadal variability in the occurrence of blocking (Kennedy et al., 2016; Brunner et al., 2017), which compounds the problem of separating externally forced changes from internal variability (Barnes et al., 2014; Shepherd, 2014). As a result, the influence of climate change on blocking remains an open question (Francis and Vavrus, 2012; Barnes, 2013; Hassanzadeh et al., 2014; Barnes and Polvani,
2015; Barnes and Screen, 2015; Francis and Vavrus, 2015; Coumou et al., 2018; Mann et al., 2018).

In order to better understand blocking and to investigate the influence of climate change, there have been significant efforts to develop methods that can automatically detect blocking in long meteorological records. Since "any attempt to identify blocked cases with certainty from an inspection of the longer available record of surface analyses would require a prohibitive expenditure of time" (Rex, 1950), blocking indices (BIs) have been developed to objectively identify blocked events (Lejenäs
and Økland, 1983; Dole and Gordon, 1983; Tibaldi and Molteni, 1990; Pelly and Hoskins, 2003). However, the multiplicity of these BIs, with a variety of thresholds for defining the area, persistence and magnitude of blocked features on different atmospheric dynamical variables, means that these methods necessarily carry the burden of somewhat subjective definitions. Notably, while previous intercomparisons of BIs show similar global climatologies, and while all indices capture many of the basic features of atmospheric blocking within their definitions, there are known regional and seasonal differences (Croci-
Maspoli et al., 2007; Barriopedro et al., 2010; Pinheiro et al., 2019). In addition, whilst spatial climatologies obtained from these BIs have been compared extensively, to the best of our knowledge there has been no direct time series comparison of the BIs beyond case study analyses such as those in Scherrer et al. (2006) and Pinheiro et al. (2019).

Other frequently used methods to study the climatology and characteristics of blocking include K-means clustering analyses to study weather regimes (Vautard, 1990; Michelangeli et al., 1995; Cassou, 2008; Ullmann et al., 2014; Strommen et al., 2019;
Fabiano et al., 2021) and an unsupervised machine learning approach called self-organizing maps (SOMs) (Skific and Francis, 2012; Horton et al., 2015; Mioduszewski et al., 2016; Gibson et al., 2017a). It has been highlighted that consistency across various methods in detecting long-term changes is a fundamental requirement to confidently identify trends (Barnes et al., 2014; Woollings et al., 2018). To the best of our knowledge, there has been no previous study that directly compared a SOM approach to other BIs.

With the advent of modern computational methods, extensive study of the available record of surface analyses to identify blocking events no longer requires a prohibitive expenditure of time. Here, we therefore define a new binary ground truth dataset

(GTD) of European blocking events across June–July–August (JJA) 1979-2019, based on a five-day threshold, reanalysis data and expert judgement. Our understanding of blocking events has been informed by the BIs and the various definitions that have been proposed, but we do not rely on any BIs for our study. This enables an independent time series comparison with

the BIs. We also compare our results to a K-means clustering approach to describing the weather regimes of the mid-latitude atmosphere. We present case studies of the prominent 2003 and 2019 European heat waves, where we show how well K-means, the BIs and SOMs describe the blocking events.

We then use SOMs to develop a new blocking index (SOM-BI, pronounced *"zombie"*). This SOM-BI method has advantages over previous BIs because it exploits all the spatial information provided in the input data and reduces the dependence on

arbitrary thresholds. It also provides a new way of studying blocking events that can more intuitively distinguish between different regimes and locations of blocking events, which the other indices are lacking. We identify the skill of different BIs by developing a binary time series identification of European blocking patterns and comparing this to our GTD using standard skill metrics discussed in section 2.6. This study is the first to define a GTD and we use it as a benchmark to compare the skill of different BIs over a region.

As a key result, we find that through comparison with three BIs used in a recent inter-comparison study (Pinheiro et al., 2019), the SOM-BI method has an improved skill at detecting regional blocking events. Since the SOM-BI method is not bound to a specific meteorological variable, we also quantify how its detection skill varies with the variable used, from geopotential height anomaly fields to potential vorticity maps. While there have been theoretical discussions on the importance of the meteorological variable used to define and identify blocking (Pelly and Hoskins, 2003; Chen et al., 2015), the variable-

dependence of skill of blocking detection methods has not been quantified before. Finally, we evaluate the performance of the SOM-BI on 41 years from the ERA5 reanalysis and 101 years of a pre-industrial control run carried out with the UK Earth System model (UKESM1-0-LL, hereafter UKESM). We identify a moderate improvement in blocking identification over the BIs for the reanalysis period and a significant improvement for the UKESM data. A key advantage is that the longer climate model simulation allow us to test the robustness of our method compared to other BIs over longer timescales, and to study the

dependence of the SOM-BI detection skill on the number of years included in the algorithm's training dataset.

Our paper is structured as follows. In section 2 and its subsections, we introduce the meteorological reanalysis and climate model data, the new GTD, the BIs, K-means, SOMs, and our new SOM-BI. In section 3, we present the main results of our analysis. We first compare the various blocking identification methods by means of the 2003 and 2019 European heat wave case studies (section 3.1), followed by an evaluation and intercomparison of the methods on ERA5 reanalysis and UKESM

climate model data (sections 3.2 and 3.3). In section 3.4, we discuss how the performance of our new SOM-BI depends on the length of the data record used to train the algorithm. In section 3.5, we test the feasibility to train SOM-BI on ERA5 data to then reliably identify blocking in climate model data, and vice versa. In section 3.6 we briefly discuss the effect of other hyperparameters on the SOM-BI skill. In section 3.7, we demonstrate how SOM-BI can be used to study trends in regional blocking patterns by applying it to ERA5 data. In section 4, we summarise and discuss our key results, and propose avenues

for future work, especially concerning the detection of blocking in climate change simulations.

## 2  Methods

### 2.1  Meteorological data

As a proxy for observed dynamical states over Europe, we used ERA5 reanalysis data from the European Centre for Medium Range Weather Forecasts (ECMWF, Hersbach et al., 2020). The pre-industrial climate model data was obtained from simulations carried out with the UK Earth System Model UKESM1-0-LL (UKESM), as part of Coupled Model Intercomparison Project Phase 6 (CMIP6, Eyring et al., 2016; Sellar et al., 2019). For ERA5, we used gridded data at a spatial resolution of $1°$ x $1°$ across 1979-2019, and created daily averages derived from 3-hourly intervals. In UKESM, we used 101 years of daily data from the pre-industrial run of the r1i1p1f2 ensemble member, across the arbitrarily defined 1960-2060 period. We used the UKESM data at the native resolution of $1.25°$ x $1.875°$ to develop the GTD plots and regridded to a $2°$ x $2°$ grid for the SOM analysis. When training and testing between the ERA5 and UKESM data (section 3.5), we also regridded the ERA5 data to a $2°$ x $2°$ grid.

For both types of datasets, we used the following common meteorological variables to characterize the dynamical state of the atmosphere at any given time: geopotential height at 500 hPa ($Z_{500}$), mean sea level pressure (MSLP), relative vorticity at 500 hPa ($\zeta_{500}$). For ERA5, we also used vertically integrated potential vorticity across 150-500 hPa (VPV), isentropic potential vorticity on 350 K and 330 K ($IPV_{350}$ and $IPV_{330}$) and potential temperature on the PV$= 2$ PVU surface ($\theta$-PV). These PV-based variables have all been used in the context of understanding atmospheric blocking (Hoskins et al., 1985; Crum and Stevens, 1988; Pelly and Hoskins, 2003) but are not available from the CMIP6 archive. The 350 K and 330 K isentropes were chosen because these intersect with the tropopause in the mid-latitude summer, as shown in Fig. 1 of Liniger and Davies (2004), and therefore represent upper-level dynamics. For the case study analyses we have also used the surface horizontal wind fields and surface temperature ($T_{surf}$).

Following Grotjahn and Zhang (2017) and Pinheiro et al. (2019), we define the anomaly fields that we study by subtracting a long-term daily mean (LTDM) from the data instead of subtracting the daily average. This is a smoothed function of the 365-day seasonal cycle across $Z_{500}$, VPV and $T_{surf}$ using the first six harmonics of their Fourier series, where the first harmonic corresponds to the mean and the fifth to a 73 day span. The purpose of this is to smooth out the daily mean fields, which can otherwise show excessive variation between neighbouring days across the seasonal cycle.

The $T_{surf}$ and $Z_{500}$ anomaly fields in ERA5 have been detrended linearly across time to remove the effect of thermodynamic warming. Following Jézéquel et al. (2017) we subtract a spatially uniform trend, so that the horizontal gradients of the field are not altered. We depart from the Jézéquel et al. (2017) method by subtracting a linear $Z_{500}$ anomaly trend instead of a cubic spline interpolation, since we assume that in the 1979-2019 time period the thermodynamic dilation of the troposphere can be approximated as linear, so removing nonlinear trends could risk removing the dynamical changes in the atmosphere that we are interested in. We also apply the same detrending approach to the pre-industrial UKESM data, to remove any minor remaining trends in the data, e.g. due to the finite spin-up time of the control simulations (Gregory et al., 2004; Nowack et al., 2017; Mansfield et al., 2020).

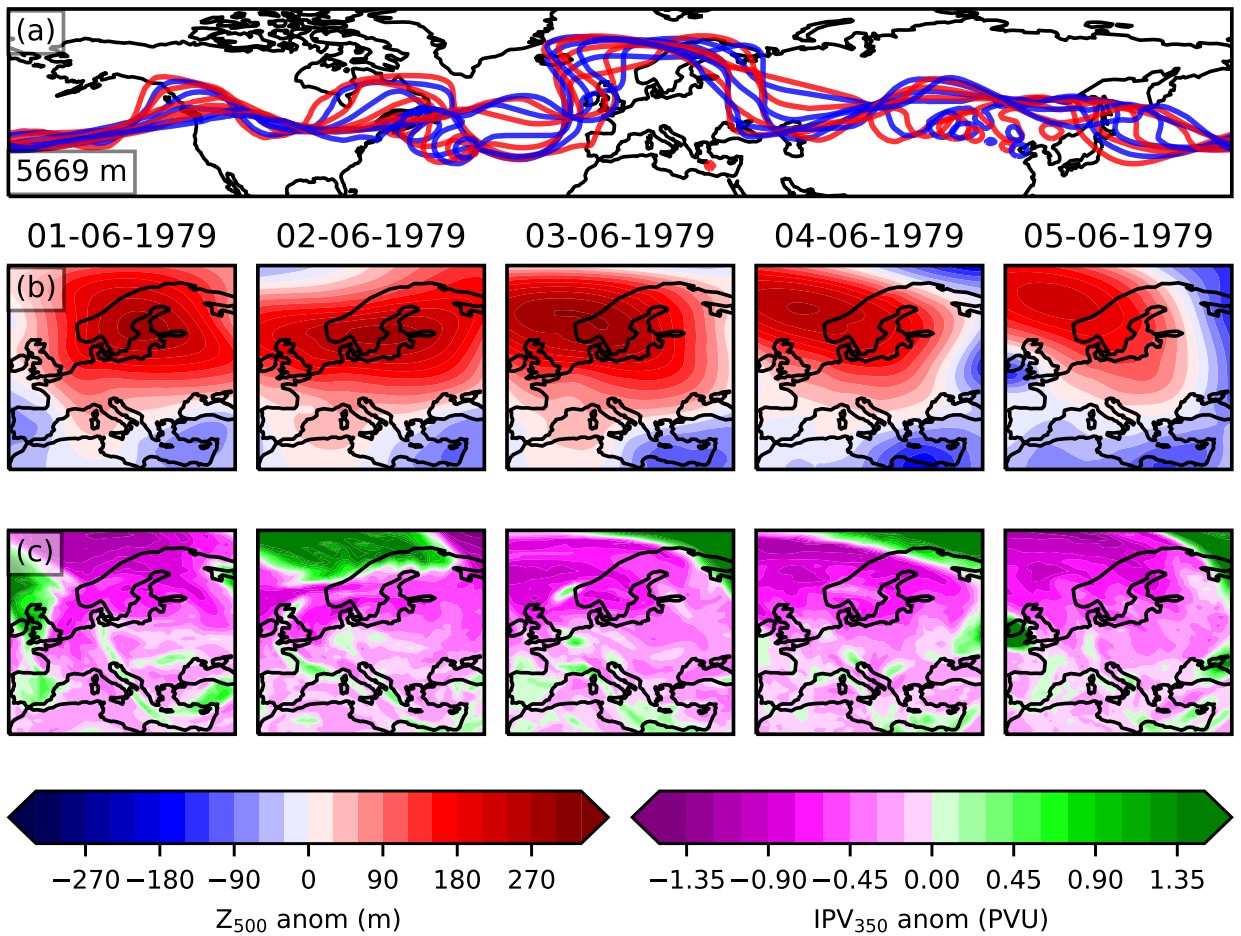

**Figure 1.** The information used to classify blocks in the ERA5 ground truth dataset (GTD). (a) shows the $Z_{500}$ contour for the averaged value across 30-70 °N, indicated in the bottom left of the panel. The red and blue colours highlight the contours at midnight and midday respectively. (b) and (c) show the $Z_{500}$ time detrended anomaly and $IPV_{350}$ anomaly for each day.

## 2.2 Creating the ground truth dataset (GTD)

In order to objectively compare the blocking indices, we develop a ground truth dataset (GTD) of blocking events in JJA Europe, here defined as 30–75 °N, 10 °W–40 °E, following IPCC AR5 definitions (Stocker et al., 2013). The northern latitude is extended to 76 °N when using data on a 2° x 2° grid. JJA Europe was chosen because of our interest in the role of atmospheric dynamics in the development of mid-latitude land heat waves. Europe is a region which has seen many recent significant heat

extremes (Christidis et al., 2014), and the role of changes in atmospheric dynamics has been a significant area of interest
(Cattiaux et al., 2013; Horton et al., 2015; Saffioti et al., 2017; Huguenin et al., 2020).

The GTD has been derived by studying every successive five-day period from 28 May 1979 to 4 September 2019, and manually identifying whether or not a blocking high persisted across any such five-day period. By including the last four days at the end of May and the first four days of September, we ensure that we capture all blocking events within the JJA period. Five days was chosen since this a typical persistence threshold for blocking indices (Verdecchia et al., 1996; Schwierz et al., 2004; Scherrer et al., 2006; Pinheiro et al., 2019), although a persistence of 7-10 days with weaker BI thresholds for amplitude and area has also been used (Rex, 1950; Lejenäs and Økland, 1983).

A diagram of the type of information analysed to label each individual day is shown in Fig. 1, for the example period 1-5 June 1979. This period was labelled as blocked, since Fig. 1a clearly shows a continuous North shift in the $Z_{500}$ contours over Europe and Fig. 1b shows a substantial positive $Z_{500}$ anomaly which persists across Northern Europe. The $IPV_{350}$ maps in Fig. 1c highlight filaments and regions where there is fast moving air. Once the total set of all 4001 consecutive five-day periods across JJA 1979-2019 has been classified, persistent *blocking events* are reconstructed to form a time series where each day is labelled as blocked or not. If a day belongs to any one of the consecutive blocked five-day periods, it is individually labelled as blocked (1), and if a given day does not belong to any of the blocked five-day periods it is labelled as not blocked (0). This creates a classification of blocking patterns for each day where each blocking event has a minimum length of five days. Blocking events longer than five days are also identified through this approach, since days that are part of any consecutive five-day blocked period are labelled as blocked. Blocking events longer than five days are then identified through a series of adjacent five-day blocked periods.

A similar approach was adopted to classify 9494 five-day periods from 101 years of JJA data in the UKESM pre-industrial control run, with an example blocked period shown in Fig. A1. As in Fig. 1, there is a clear quasi-stationary high centered on a region slightly north of the UK. This is indicated by the $Z_{500}$ contours which show a significant northward protrusion over this region, and by the substantial $Z_{500}$ anomaly across all panels in Fig. 1b. Since PV is not available in CMIP6 data and the physical variables used to derive PV are not available at sufficiently high vertical resolution, we instead show the MSLP anomaly field in Fig. A1c, which also indicates a high pressure region consistent with Figs. A1a and A1b.

## 2.3 Blocking Indices (BIs)

One way of describing atmospheric flow and investigating trends in atmospheric dynamics is by using proxy indices such as those used to classify if a blocking event is occurring. There are many blocking indices (BIs) that have been used to create a blocking climatology, and these have been rigorously compared (Barriopedro et al., 2010; Pinheiro et al., 2019). Some BIs are based on measuring persistent anomalies of a relevant pressure field in a particular location. This builds on the pioneering work of Elliot and Smith (1949), who identified events of persistent sea level pressure (SLP) anomalies above a particular threshold. This approach was extended by Dole and Gordon (1983) who investigated persistent anomalies in the $Z_{500}$ field. A similar approach was taken by Schwierz et al. (2004) who identified anomalies in the vertically averaged potential vorticity field (VPV), averaged over 150-500 hPa. This approach was inspired by the work of Pelly and Hoskins (2003), who defined

blocking as the negative latitudinal potential temperature gradient on the dynamical tropopause. By taking a vertical average of the potential vorticity field from the mid-troposphere to the lower stratosphere, Schwierz et al. (2004) formulate a 3-D dynamically based index.

Another common approach to studying blocking trends is to use the absolute gradient of $Z_{500}$ across fixed latitudes. This was first developed in Lejenäs and Økland (1983) and refined in a commonly applied form by Tibaldi and Molteni (1990). This definition focuses on blocking events as persistent anticyclones that reverse the $Z_{500}$ gradient. The method has been adopted widely, refined (Diao et al., 2006; Barriopedro et al., 2010), and extended to a range of latitudes (Scherrer et al., 2006).

All of these methods have been further developed by Pinheiro et al. (2019) who applied four thresholds for each blocking index: the magnitude of the anomaly, the persistence of the blocking event (minimum five days), a minimum area over which the anomaly takes place and an overlap criterion which measures if there is continuity across the blocked region between different days (an overlap of the blocked contours). We adopt their thresholds and as such study the three indices compared in Pinheiro et al. (2019) including their modifications:

- **AGP** - the geopotential height gradient method, which is the Tibaldi and Molteni (1990) index as adapted by Scherrer et al. (2006) to construct a two-dimensional field of geopotential height gradients

- **DG83** - the Dole and Gordon (1983) method of investigating positive geopotential height anomalies

- **S04** - the Schwierz et al. (2004) method of identifying persistent anomalies in the potential vorticity field (VPV) averaged over 150-500 hPa (VPV).

We refer the reader to section 2.2 in Pinheiro et al. (2019) for a detailed discussion of these methods and their associated thresholds. However, our analysis differs from the methodology outlined by Pinheiro et al. (2019) in three ways, reflecting the fact that our study is regional and seasonal instead of global. Firstly, we apply all thresholds defined by Pinheiro et al. (2019) only to those grid cells within the region of study so that we exclude events that are on the edges of the domain. Such events would be considered blocking events if the domain studied was extended. Secondly, Pinheiro et al. (2019) applied a spatial smoothing to their global threshold field, which defines the minimum threshold for each grid cell to be blocked. Although we have applied the LTDM smoothing of the seasonal cycle (which we subtract from variables to calculate field anomalies, section 2.1) and we also use a spatially varying threshold field, we have not applied this spatial smoothing to our threshold field. We found that the resulting blocking climatologies shown in Fig. A4 are broadly consistent with those presented in Fig. 6 of Pinheiro et al. (2019), underlining that this regional use of the BIs is still valid. Finally, to remove the well-known problem of the AGP index identifying anomalous blocking events associated with the subtropical high in summer (Davini et al., 2012), we adopt the extra threshold of the AGP index from Woollings et al. (2018). The subtropical high feature was not observed in UKESM over Europe, since the zonal gradients have a smaller magnitude, so the standard AGP index is used for UKESM.

We note that more indices have been proposed, including hybrid approaches combining the AGP and DG83 indices (Barriopedro et al., 2010; Dunn-Sigouin et al., 2013; Woollings et al., 2018), the PV-$\theta$ approach developed by Pelly and Hoskins (2003) and the finite amplitude wave activity (FAWA) method (Huang and Nakamura, 2015). K-means clustering analysis (Diday and Simon, 1980) has also been extensively used to study the Euro-Atlantic midlatitude variability and to identify weather

regimes (Vautard, 1990; Michelangeli et al., 1995; Cassou, 2008; Ullmann et al., 2014; Strommen et al., 2019; Fabiano et al., 2021). However, with the three BI methods included here in addition to the SOM-BI and K-means comparison in the case studies, we expect to see results that are sufficiently representative of the range of blocking detection methods available, and to be able to highlight their most important similarities and discrepancies.

## 2.4 Self-organizing map (SOM)

The fourth method we used to investigate trends in atmospheric circulation regimes in European summer is self-organizing map cluster analysis (SOM; Kohonen, 1982). This is an increasingly popular unsupervised machine learning technique in synoptic meteorology to learn representative patterns of weather regimes and to investigate their trends (Hewitson and Crane, 2002; Liu and Weisberg, 2005; Huth et al., 2008; Sheridan and Lee, 2011; Johnson, 2013; Horton et al., 2015; Xu et al., 2016; Singh et al., 2016; Diffenbaugh et al., 2017; Sánchez-Benítez et al., 2019). In our context here, the SOM algorithm is trained with daily spatial maps of dynamical states of the atmosphere above Europe, as for example characterized by maps of geopotential height anomalies (Fig. 1b), potential vorticity (Fig. 1c) or sea level pressure (Fig. A1b). By iteratively cycling through all samples of such meteorological maps, the algorithm learns representative patterns of atmospheric dynamical states, which are referred to as "SOM nodes".

First, the number of nodes is specified and the SOM is initialised either with random values or with principal component analysis patterns. Then for each day from the input field, the Euclidean distance between that daily meteorological pattern and each node pattern is calculated. The node with the smallest Euclidean distance to the sample day is known as the best matching unit (BMU) for that day. Then the BMU pattern is updated to shift towards the pattern of the sample day. The neighbouring SOM nodes (on the grid of SOM nodes) are also updated to shift towards the sample day according to a Gaussian neighbourhood function. For each cycle of iterations through all training samples, the updates tend to become smaller as the SOM nodes converge towards a representative pattern of atmospheric dynamical states. A decay function on the updates is additionally applied, which ensures convergence. Finally, a stable SOM is obtained with a set of nodes that each provide a representative composite of circulation patterns, arranged according to their similarity on a row-column grid (i.e. the map). A diagram of the training procedure is shown in Fig. 2. The number of nodes to be learned by the algorithm, or in other words the number of representative weather patterns one aims to learn for a particular meteorological problem, is chosen by the user. In section 3.3, we show how the SOM-BI performance depends on the number of nodes and how this provides an objective criterion to select this number.

SOMs are of particular relevance in atmospheric science because they maintain the topological properties of the input space. Once optimised, each node pattern represents a possible state of the atmosphere, and the nodes are arranged in order of similarity, thus representing a continuum of atmospheric states. This contrasts with other methods of dimension reduction such as principal component analysis, where the identified patterns are orthogonal. Such purely mathematical representations are typically less meaningful from a physical point of view, whilst each SOM node maintains physical significance as it can closely resemble actual atmospheric states found in meteorological data, with the nodes on the row-column grid representing

## Initial SOM (x10$^{-3}$)

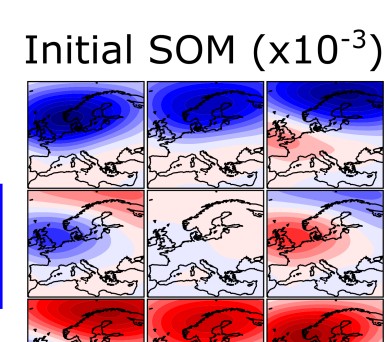

1) Choose the number of nodes and initialize the SOM using principal component analysis

Example SOM before new sample day

Sample day    BMU node    Updated BMU

Updated SOM

2) For each day from the input data, identify the node in the SOM which is the closest match to the data sample, known as the best matching unit (BMU)

3) The position of the BMU and its neighbouring nodes is shifted towards the data sample

## Final SOM

−270 −180 −90    0    90   180   270
Z$_{500}$ anom (m)

4) Repeat steps 2 and 3 until a stable SOM is obtained that provides a representative set of circulation patterns

**Figure 2.** The self-organizing map algorithm. Shown using a 3x3 node SOM with ERA5 Z$_{500}$ JJA 1979-2019. The PCA-initialised SOM pattern (step 1) has a much larger amplitude so has been multiplied by $10^{-3}$ for visualisation purposes. The BMU refers to the best-matching unit, the SOM node which most closely matches the sample day.

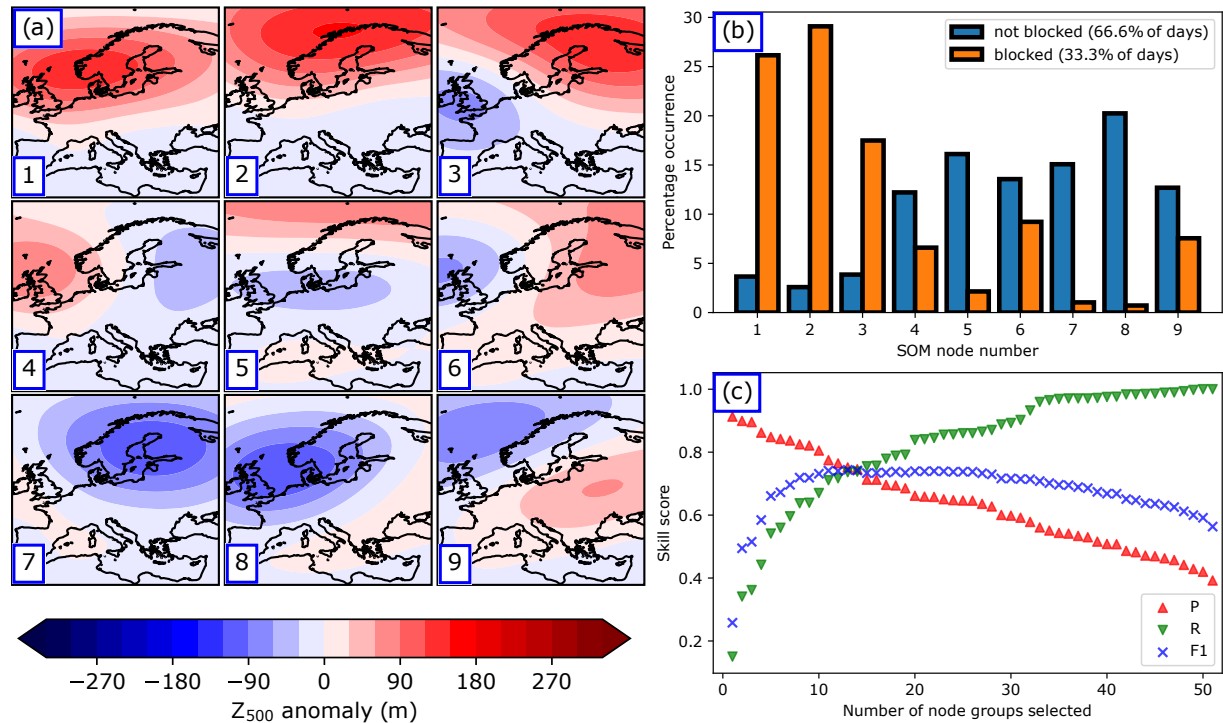

**Figure 3.** The SOM blocking index (SOM-BI). (a) The trained 3x3 SOM for $Z_{500}$ time-detrended anomaly. (b) Normalised histograms showing the distributions of occurrence of BMUs for the days identified as blocked or non-blocked within the GTD. (c) The SOM-BI optimisation of the set of node groups against three different skill scores (precision (P), recall (R) and F1 score) that are associated with the GTD blocking events.

smooth transitions across those possible atmospheric states (see the similarity of neighbouring nodes in the final SOM grid in Fig. 2). We have implemented the SOM algorithm using the somoclu Python package (Wittek et al., 2017).

This property of SOMs is also the distinguishing feature between SOMs and K-means clustering. In the case of K-means clustering, each node is updated at each iteration independently and no neighbourhood function is applied. K-means tries to maximize differences between the centroids such that it does not learn a topology. This difference between K-means and SOMs is minor for low node numbers, since the sharp differences in spatial patterns are imposed on the SOMs and the neighbourhood function has a limited effect. For larger node numbers, the SOM topology becomes smoother and the K-means centroids remain distinct rather than representing a continuum of states, whereas a continuum is a more realistic reflection of the actual atmosphere (Skific and Francis, 2012). A comparison between SOMs and K-means analysis for 4 and 20 node/cluster numbers is shown in Fig. A5.

## 2.5 The self-organizing map blocking index (SOM-BI)

Once we have created the GTD, this can be used to develop a new BI using SOM analysis. For a given variable from the ERA5 dataset, we can specify a node number and arrangement of nodes (number of rows and columns, Fig. 2) and then learn the corresponding SOM nodes from that data. Figure 3a shows the trained pattern for $Z_{500}$ anomalies in ERA5 28 May-4 Sep 1979-2019 for 9 nodes arranged in a 3x3 grid. Since each day in the dataset has been matched to a BMU, we can identify which nodes are associated with blocked days according to our GTD. Figure 3b compares the histograms of those nodes which are and are not associated with the GTD blocking events. As expected, the three nodes with large positive $Z_{500}$ anomalies (nodes 1, 2 and 3) are most closely associated with blocking events, and the nodes with large negative $Z_{500}$ anomalies (nodes 7 and 8) are rarely associated with blocking events. However, nodes 1, 2 and 3 still occur on 15% of non-blocked days, and 28% of the blocked days are also matched with one of the other six nodes, including 3% of blocked days matched with nodes 7 and 8. This tells us that while the SOM nodes can indicate the occurrence of blocked events, there is no node or single combination of nodes that can be consistently identified with blocking events with high skill.

However, from every five-day period within the GTD, we can identify an associated "group" of nodes. For example, a five-day period can be associated with nodes 1 and 4 (any arrangement of nodes 1 and 4 across five days), and this would mean that [1,4] is the associated group of nodes for that five-day period. Since each five-day period has been classified either as blocked or not blocked, it raises the possibility that a set of such groups can be more specifically associated with blocking. We identify the optimal set of node groups associated with blocking by ordering the list of all possible node groups (e.g. [1,2,3], [1,4], [1], [1,2,3,4,6] etc) from the node groups that have the highest to lowest precision (P) at identifying blocking events.

## 2.6 Classification skill measures

Fig. 3 (c) shows the binary classification skill according to the measures of precision, recall and F1 score when applying the 9-node SOM-BI to ERA5 data. The three skill measures are shown for consecutive cases where we successively add node groups as described above in order from highest to lower precision to the set of groups that we associate with blocking. In other words, once a new group has been added to the set of groups, this new group will define a series of blocked periods within our SOM-BI approach. Precision (P) is defined as the ratio of true positives to total detected positives. For example, a precision of 0.8 indicates that 80% of the events identified by a method are true positives and the remaining 20% are false positives. Recall (R) is the number of true positives divided by the total number of actual events. A recall of 0.8 indicates that 80% of all total blocking events are captured by the classification method, but 20% are false negatives. A higher recall is typically associated with a loss in precision, as identifying more events also means that one typically identifies more false positive events. Therefore, a careful balance between precision and recall is usually sought after. One widely used skill metric to achieve this balance is the F1 score, which is the harmonic mean of precision and recall:

$$F1 = \frac{2 \cdot P \cdot R}{P + R} \tag{1}$$

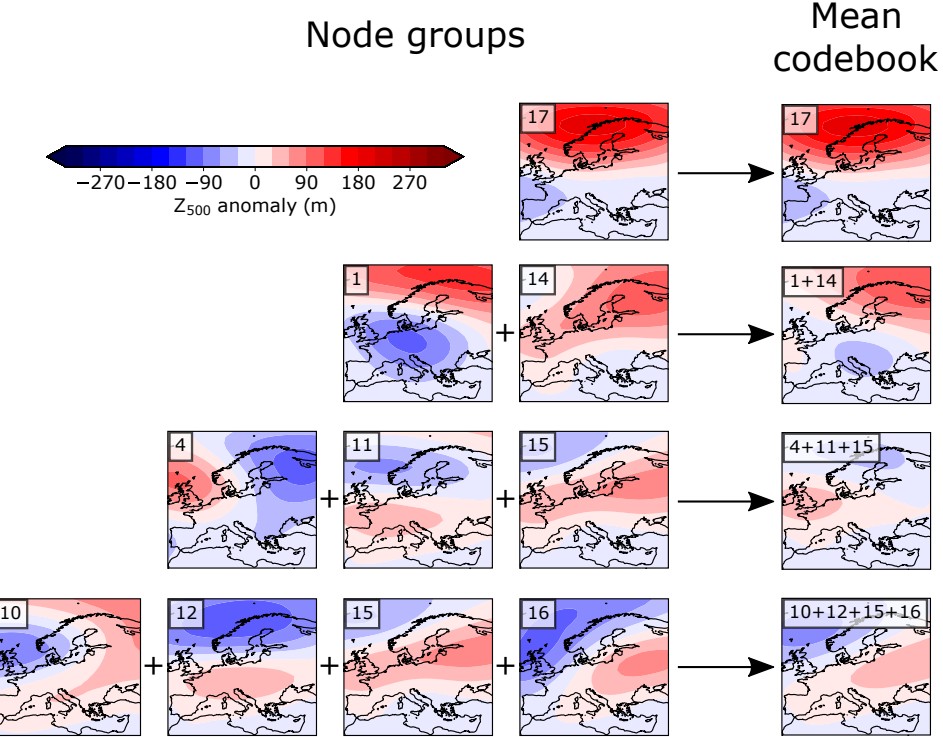

**Figure 4.** Examples of four blocked node groups identified by the SOM-BI described in section 2.5 averaged to form mean codebooks. Shown here for $Z_{500}$ in the optimised case of 20 nodes. There are 114 blocked node groups in total.

which can vary between 0 (worst case, low detection skill) and 1 (best score). If either P or R are low, the F1 score tends towards 0, thus indicating low detection skill in at least one of the two measures. For example, if there is a small number of node groups selected in the SOM-BI, the precision is very high but the recall is very low - a small number of blocking events is well described but many blocking events are missed by the classification. When a larger number of node groups with a decreasing precision is included, then precision decreases and recall increases; more events are described but there is also a higher proportion of false positives. For the 3x3 SOM learned from ERA5 data, the node group with the highest precision is [1], with P = 0.91 and R = 0.15, followed by [2] with P = 0.89 and R = 0.19 and [1, 2, 6] with P = 0.87 and R = 0.03. If only one node group is included in the set (e.g. [1] or [1, 2, 6]), there is a high P and low R, but as more node groups are added to the set of node groups (e.g. [1], [2]; then [1], [2], [1, 2, 6]), P decreases but R increases. We identify the optimal set of node groups by the value which maximises the F1 score (Fig. 3c). We perform this classification for a range of node numbers and meteorological variables to identify an optimal performance in section 3.3.

## 2.7 SOM-BI application

Once an optimal set of node groups has been identified, these can be used to classify days as blocked or not blocked. This creates a time series of blocking events but it does not produce a spatial climatology. To develop a spatial climatology for the SOM-BI, we use the BIs described in 2.3 across the days that are identified as blocked by the SOM-BI.

A key advantage of the SOM-BI is that it identifies distinct types of regional blocking events, since each blocked node group within the set of node groups is associated with a set of blocking events. In the example shown in Fig. 3, 14 node groups are associated with blocking at the intersection of precision and recall, which therefore identifies 14 possible distinct types of blocking. For example, the node group [1] describes broad NW European events, [2] describes Scandinavian blocking, and [1, 2, 6] describes a more variable set of blocking patterns that are broadly associated with NE Europe.

To aid in our interpretation of these node groups, we calculate the mean of their node codebooks, i.e. the mean of the spatial patterns of the nodes in each node group, which in turn also characterize the corresponding blocking patterns. This forms "mean codebooks" for each node group. Figure 4 shows four examples of such node groups associated with blocking from ERA5 $Z_{500}$ for the case of 20 nodes - the optimum number of nodes for this case (cf. Fig. 7a). These four node groups are chosen since they illustrate the variety of nodes and numbers of nodes present across the set of blocked node groups, and also represent a variety of spatial patterns in blocking (N, NW, W and E). In section 3.7, these mean codebooks are applied to identify distinct categories of blocking and to study their historical trends in ERA5.

## 3 Results

### 3.1 Case study analyses

We compare the blocking identification methods (i.e. SOMs/SOM-BI, the three conventional BIs, and K-means) for two examples of well-known 2003 and 2019 European heat waves that were linked to blocking states of the atmosphere (Figs. 5 and 6). In addition, we study two blocking events from UKESM, to investigate how blocking events are described in the climate model. From the 101 years investigated in the pre-industrial control run we have found the largest extent of heat extremes to occur in an extended heat wave shown in Appendix Fig. A2. This is contrasted with Fig. A3, which shows the end of a blocking event and a weaker transitory anticyclone. Both UKESM events are discussed further in Appendix A.

The 2003 European heat wave was a record-breaking heat wave that had significant societal impacts (Robine et al., 2008) and was shown to have been made at least twice as likely due to anthropogenic climate change (Stott et al., 2004). According to climate change projections, such heat waves will become commonplace by the 2040s irrespective of future emissions scenarios (Christidis et al., 2014). The most extreme temperatures during this heat wave were recorded from the 6-12 August, where the peak temperature recorded was in Southern France at 41°C. Black et al. (2004) reports that atmospheric flow anomalies were recorded in early August, although there was a relatively weak signature of blocking. The 2003 heat wave remained the European temperature record until 2019, when surface temperatures of 46°C were observed in central France. The 2019 heat

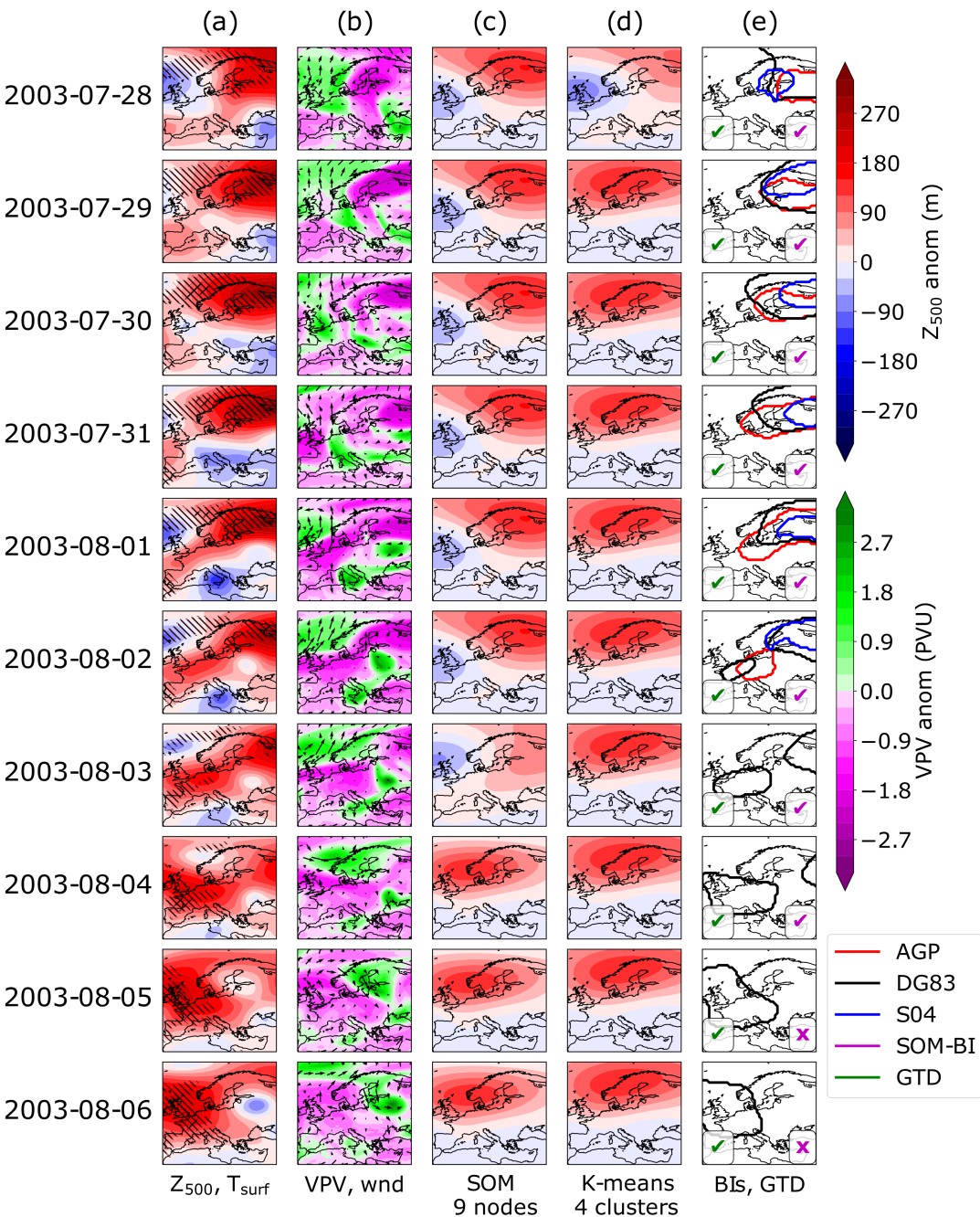

**Figure 5.** The 2003 European heat wave. (a) shows the detrended 500 hPa geopotential height anomaly for each day. Left (right) hatching indicates where the local surface temperature exceeds the 90th (99th) percentile for the detrended 2 m temperature. (b) shows the potential vorticity anomaly vertically averaged across 150-500 hPa, with arrows showing the 10-m wind field. (c) shows the corresponding SOM pattern for $Z_{500}$ anomalies from 9 nodes. (d) similarly shows the corresponding K-means centroid for 4 clusters. (e) shows the contours identified as blocked in this region in the AGP (red), DG83 (black) and S04 (blue) indices. A green (magenta) tick or cross indicates if the GTD (SOM-BI) identifies the day as blocked or not.

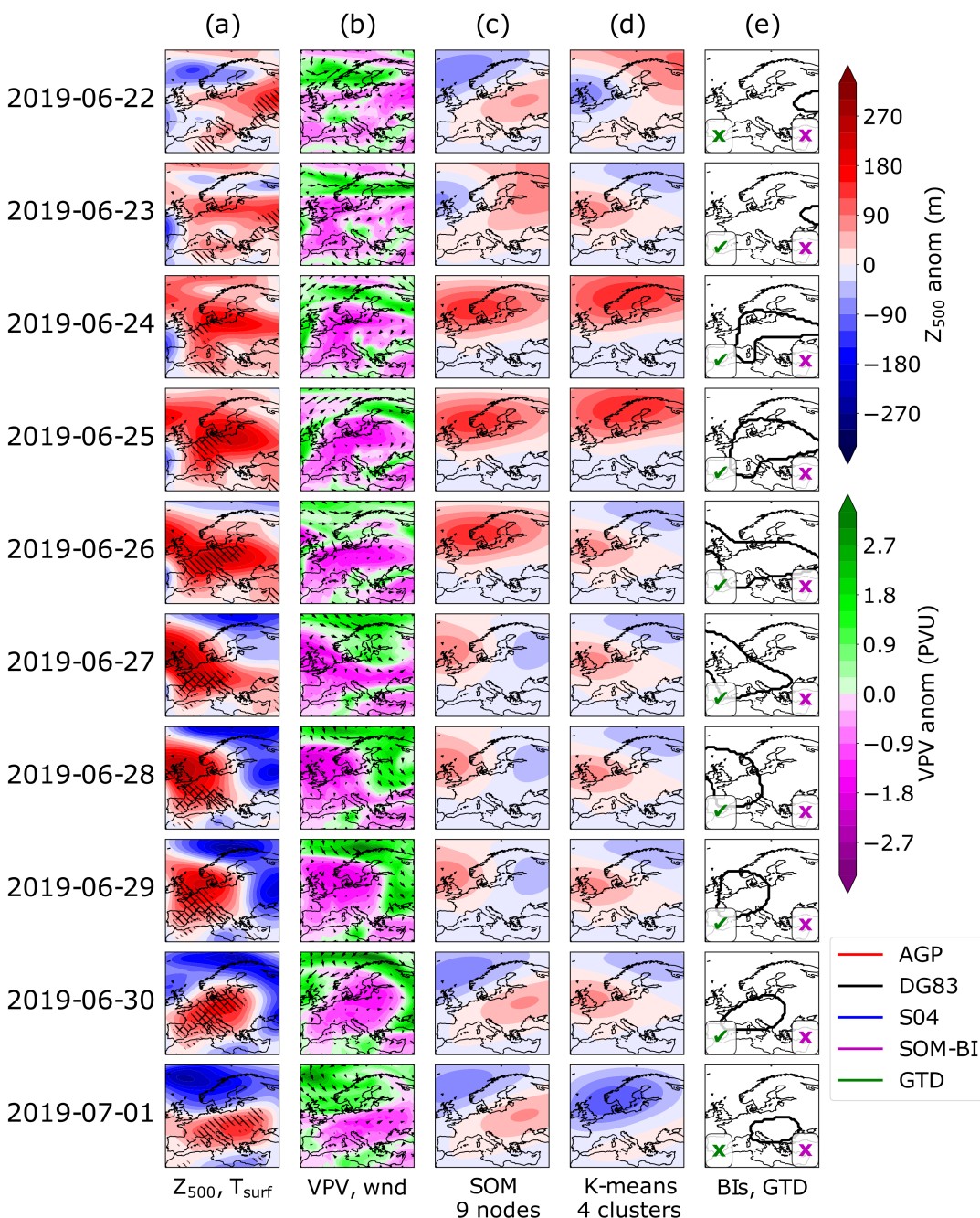

**Figure 6.** As in Fig. 5 but for the 2019 European heat wave.

wave was concurrent with persistent hot air that originated in North Africa (the so-called "Saharan heat bubble"), which was sustained by an omega block centered on Western Europe (Mitchell et al., 2019).

Figs. 5a and 6a show daily maps of detrended $Z_{500}$ anomalies for the two events, the field used by the DG83 index to identify blocking events. The hatching indicates detrended surface temperature extremes at the 90th and 99th percentile. It can be seen that across all cases there are significant positive $Z_{500}$ anomalies which are associated with temperature extremes. Figures 5b and 6b show the vertically averaged potential vorticity (VPV) field, used by the S04 index to identify blocking, and also the 10 m winds. The VPV field is consistently anti-correlated with the $Z_{500}$ field, and significant negative anomalies in the VPV

field tend to be associated with stationary surface winds, particularly across 26-29 June 2019. Figures 5c and 6c show the BMU SOM pattern for the case of 9 nodes for detrended $Z_{500}$ anomaly fields. Whilst the SOM nodes clearly track the features shown in the $Z_{500}$ maps, a range of BMUs are identified in both case studies even though there is a consistent extreme weather event across these time periods. In the 2003 case study in ERA5, three SOM BMUs and four transitions between BMUs are shown in Fig. 5c. These all show positive $Z_{500}$ anomalies in the Northern part of the domain, even though the meteorological situation

varies meridionally more than zonally, particularly across 4-9 August 2003. An even greater variety of BMUs is observed in the 2019 case, where four nodes and four transitions between SOM nodes are shown in Fig. 6c.

This creates a difficulty of interpretation - whilst the SOM can identify the best matching spatial pattern of $Z_{500}$ anomalies, these particular SOM patterns do not correspond to circulation regimes as conventionally understood, since even minor shifts in the domain (such as the change from the 2-3 August 2003) can cause the corresponding pattern to shift. The frequency of

these shifts and sensitivity of the SOM is dependent both on the number of nodes chosen and the domain size. Smaller domains with fewer SOMs show more consistency in the synoptic weather patterns, but when these are sufficiently reduced (such as for four SOMs over the Mediterranean), the SOMs become less distinguishable and lose even more of their explanatory power to represent meaningful pattern variations across the domain (not shown). Overall, the fact that several SOM nodes occur during the case study blocking events shows that individual SOM patterns will not be able to consistently identify blocking events with

high precision or recall, contrary to how SOMs are typically used in many applications in the literature. However, well-defined groups of nodes, as we will show below, can indeed achieve this task and can thus be used for the purpose of our new SOM-BI.

Figures 5d and 6d show a K-means clustering analysis using $Z_{500}$ anomaly fields for the case of 4 centroids. As described in section 2.4, the case of K-means with 4 centroids produces a similar set of weather regimes to SOMs with 4 nodes. Consequently, the K-means analysis exhibits a similar behaviour to the SOMs discussed above but distinguishing between fewer

weather regimes. One weather regime indicating Scandinavian blocking consistently represents the 2003 European heat wave across Fig. 5d, but the Westward shift of the high pressure centre from Scandinavia on 31 July to the UK on 8 August 2003 is not described by 4 centroids. For the 2019 heat wave in Fig. 6d, all four weather regimes are represented, and the blocked period is primarily associated with a mixed weather regime. This shows that the 2019 case is also not described well by K-means clustering.

Figs. 5e and 6e show the contours demarking blocked regions as identified by the three different BIs. A tick or cross in the bottom left and right corners indicates whether the day was identified as blocked or not in the GTD or the SOM-BI. For the SOM-BI labelling, $Z_{500}$ 20 SOM nodes are chosen on the basis of the optimisation of the SOM-BI in Fig. 7a. Across all

| Dataset | Method | Days blocked | Precision | Recall | F1 | F1 wrt AGP | F1 wrt DG83 | F1 wrt S04 | F1 wrt SOM-BI |
|---|---|---|---|---|---|---|---|---|---|
| ERA5 | GTD | 33.4% | 1 | 1 | 1 | **0.56** | **0.73** | **0.19** | **0.74** |
| ERA5 | AGP | 19.5% | 0.76 | 0.44 | 0.56 | 1 | 0.55 | 0.22 | 0.51 |
| ERA5 | DG83 | 34.3% | 0.72 | 0.75 | 0.73 | 0.55 | 1 | 0.19 | 0.69 |
| ERA5 | S04 | 5.3% | 0.69 | 0.11 | 0.19 | 0.22 | 0.19 | 1 | 0.15 |
| ERA5 | SOM-BI | 34.6% | 0.73 | 0.75 | 0.74 | 0.51 | 0.69 | 0.15 | 1 |
| ERA5 | BLO | 100% | 0.33 | 1 | 0.50 | 0.33 | 0.51 | 0.10 | 0.51 |
| ERA5 | RND | 33.4% | 0.33 | 0.33 | 0.33 | 0.25 | 0.34 | 0.09 | 0.34 |
| UKESM | GTD | 29.0% | 1 | 1 | 1 | **0.34** | **0.60** | - | **0.71** |
| UKESM | AGP | 20.8% | 0.41 | 0.29 | 0.34 | 1 | 0.29 | - | 0.28 |
| UKESM | DG83 | 14.5% | 0.90 | 0.45 | 0.60 | 0.29 | 1 | - | 0.55 |
| UKESM | SOM-BI | 29.6% | 0.70 | 0.72 | 0.71 | 0.28 | 0.55 | - | 1 |
| UKESM | BLO | 100% | 0.29 | 1 | 0.45 | 0.34 | 0.25 | - | 0.46 |
| UKESM | RND | 29.0% | 0.29 | 0.29 | 0.29 | 0.24 | 0.19 | - | 0.29 |

**Table 1.** A comparison of skill scores of the original BIs and the new SOM-BI against the GTD for ERA5 1979-2018 and UKESM for JJA over the European domain. Where not indicated the skill scores are measured with respect to the relevant GTD. "BLO" indicates the skill score for the trivial case of every day labelled as blocked, and "RND" where a random allocation of blocked days has occurred with the same proportion of blocked days as the GTD.

case studies the DG83 index clearly tracks regions of positive $Z_{500}$ anomalies. The S04 and AGP indices also track the same feature in the 2003 heat wave until 3 August 2003, but does not identify any blocking associated with the 2019 heat wave. The
SOM-BI describes the initial period of the 2003 heat wave well, although it does not capture the Western European blocking event during the peak period of extreme temperature across 6-12 August. The SOM-BI also does not capture the 2019 blocking pattern coincident with the 2019 heat wave. This is because the SOM nodes are too variable over the 2019 event such that the set of nodes which best match the $Z_{500}$ anomaly fields are not generally associated with blocking. For example, the SOM nodes across 27-30 June 2019 indicate mixed patterns which do not obviously correspond to blocking over a consistent area
(the positive maxima shift from the British Isles to Eastern Europe within a day). This lack of pattern consistency is mostly the result of an unfortunate balance between the positive and negative $Z_{500}$ anomalies on these days, where the latter play a major role in the allocation of the BMU during this period. We discuss the possibility of ignoring negative anomalies in our assignments of the BMUs in section 3.6, but found that this modification does not improve the SOM-BI performance overall. In summary, there are blocking events such as these that will also not be described well by the new SOM-BI, but as we show
below the SOM-BI performs as good as or even better than many conventional BIs in most cases.

## 3.2 Blocking index comparison in ERA5 and UKESM with GTD

A climatological comparison of the BIs over JJA Europe confirms what has been discussed in the case study analyses above, and is consistent with the results of Pinheiro et al. (2019), which are broadly consistent with other BI climatologies. We show the spatial distribution of blocking climatologies according to three conventional blocking indices in Fig. A4. Where our analysis substantially differs from the literature is in our regional approach and consideration of direct time series comparisons among the BIs as well as to our new SOM-BI. We do not explicitly consider the time-averaged climatological distributions of blocking events over Europe (as shown in Fig. A4). For our comparison, we first apply all BIs to the historical ERA5 data over the European domain. Each day for each BI is labelled as blocked if a blocking event has been identified within the European sector and persists for at least five days. A blocking event is not identified if the thresholds for amplitude, persistence, area and overlap discussed in section 2.3 are not met within the European domain. This results in a binary dataset for each BI that identifies periods of at least five consecutive days where blocking patterns exist within the European sector. These binary BI data sets we then compare to our manually labelled GTD.

Table 1 compares the precision, recall and F1 scores of these BIs and our new SOM-BI against the GTD for this domain-based comparison in both ERA5 and UKESM. We further compare the time series of blocking classifications among the BIs themselves to quantify how consistent the BIs are with each other. The key results are underlined. In both ERA5 and UKESM, the best blocking index is the SOM-BI, with a F1 score of 0.74 in ERA5 and 0.71 in UKESM. All of the indices consistently perform worse in UKESM than in ERA5. This is because blocking is less frequent in the model and several of those blocking patterns identified in UKESM are less distinct (Fig. A3). This is probably associated with mean biases in the representation of $Z_{500}$ that have been observed across several climate models (Scaife et al., 2010; Schaller et al., 2018). The DG83 index performs almost as well as the SOM-BI in ERA5 with an F1 score of 0.73, but there is a significant reduction in performance to 0.60 when applied to UKESM data. The AGP index in turn shows an even weaker skill than DG83 in both reanalysis and model, with a larger drop in skill from 0.56 to 0.34 in ERA5 and UKESM, respectively. The fact that SOM-BI still shows a relatively good score for UKESM of 0.71 suggests that the SOM-BI can be particularly useful in studying regional blocking in climate models. In particular, since a model ensemble may exhibit a variety of intensities of blocking, the SOM-BI would be able to overcome the limitations of BIs, where (particularly in the case of AGP) thresholds are defined with respect to the observational record. Since the anomalous flow patterns associated with blocking will be more consistent across datasets, the SOM-BI can identify blocking events across a model ensemble with greater accuracy. The consistent skill of the SOM-BI across both ERA5 and UKESM has been further verified by swapping the training and test datasets between each dataset, as described in section 3.5.

A case where every day in Europe is labelled as blocked ("BLO") is also compared, which represents the case of perfect recall (=1) but a low precision. This case gives an F1 score of 0.53 for the GTD for ERA5 and 0.45 for the GTD of UKESM, and provides a useful benchmark of basic performance. Surprisingly, the AGP index only performs marginally better than BLO for ERA5, and performs worse in the UKESM case. Whilst S04 has a higher precision than BLO, because the recall is so low the total F1 score is much lower (0.19). Finally, we compare a random labelling of blocked and non-blocked days, where the

proportion of blocked days is equal to that of the GTD ("RND"). This gives an equal precision and recall because the number of true positives is equal to the number of false negatives. The F1 score of RND still exceeds that of S04, with 0.33 for ERA5 and 0.29 for UKESM, and is comparable to the F1 score of AGP in UKESM.

### 3.3  SOM-BI skill dependence on the choice of SOM node number and the meteorological variable

The key hyperparameter in the SOM-BI is the number of nodes ($k$), which here is similar to identifying the optimal number
of circulation patterns required to effectively classify European summer weather regimes. In addition, there are a number of meteorological variables from which we could learn the SOM patterns, which in turn will also influence the skill of our SOM-BI method. The dependency of the skill of our BI on these two factors is quantified in the following. Figures 7 and 8 show how precision (P), recall (R) and F1 score depend on $k$ and the meteorological variable in ERA5 and UKESM, respectively. Specifically, we compare the skill metrics for cases where we learn the SOM nodes from $Z_{500}$, MSLP and $\zeta_{500}$ anomalies. For
ERA5, we additionally consider four PV-related variables (VPV, $\theta$-PV, $IPV_{350}$ and $IPV_{330}$) shown in Fig. 7d-g.

Another hyperparameter related to the number of nodes is the row x column ($n x m$) arrangement of nodes. For example, 16 nodes can be arranged as 16x1, 4x4, 8x2, 2x8 or 1x16. These different arrangements affect the topology of the SOM, the initialization of the nodes and which nodes are counted in the neighbourhood of other nodes during the update process of the SOM (Fig. 2). For each $k$ in Figs. 7-9 we have used the arrangement of nodes that maximises the average number of nearest
neighbours between each node (e.g. using 4x4 nodes for $k = 16$). This approach maximally exploits the SOM topology. We have also used $n \geqslant m$ (for example using a 9x2 arrangement instead of a 2x9 arrangement of nodes for $k = 18$) to preferentially arrange the SOM topology zonally across the domain rather than meridionally. This is done because there is greater variability in the occurrence of blocking patterns zonally than meridionally across Europe (Fig. A4).

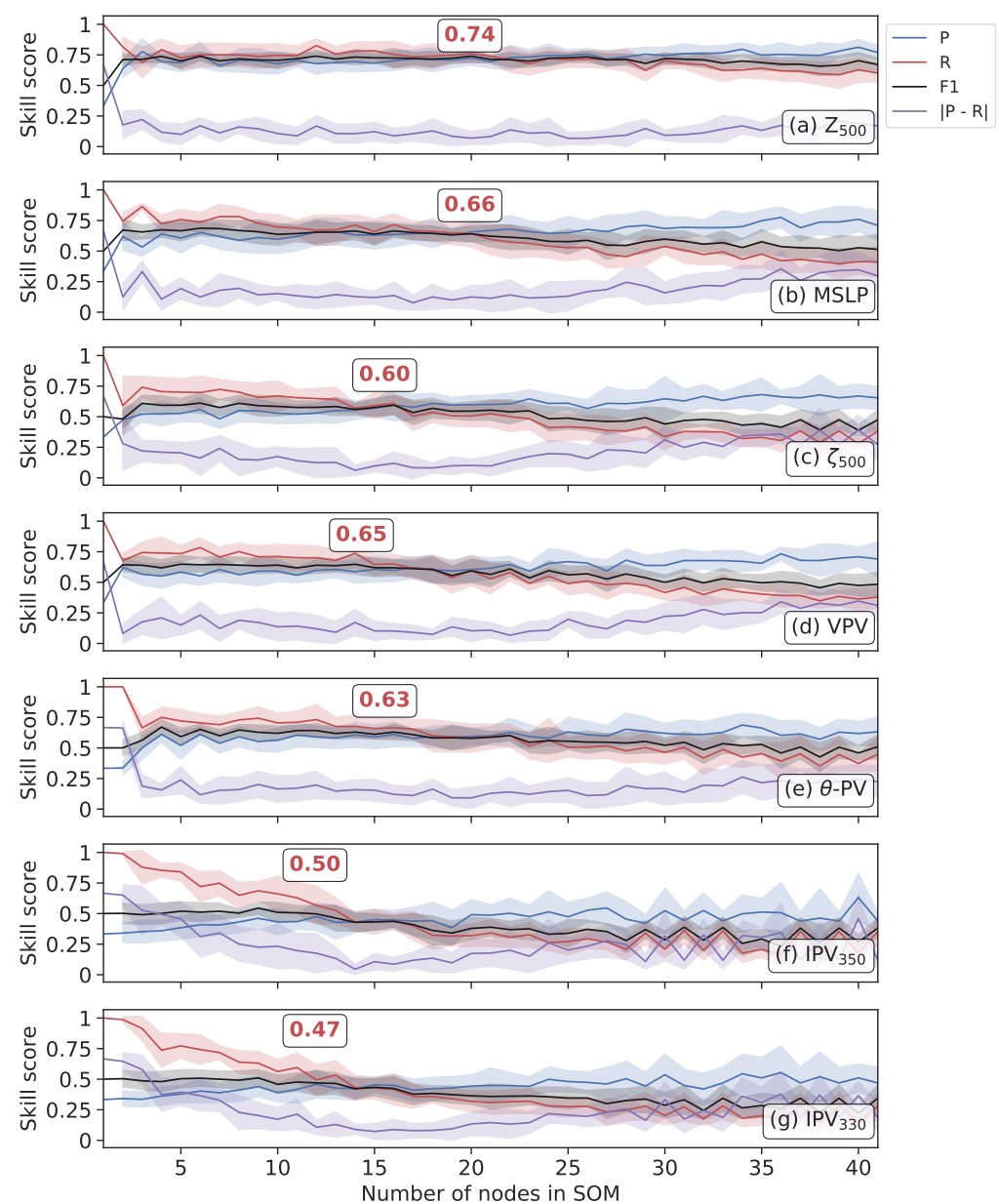

**Figure 7.** A comparison of the performance of the SOM blocking index for seven variables in the ERA5 1979-2019 historical period with a varying number of nodes in the SOM. Precision (P), recall (R) and F1 scores are calculated, and the absolute difference between precision and recall is also shown. Error bands show the standard deviation ($\pm 1\sigma$) for 10-fold cross-validation. The red number inset into each panel shows the optimal F1 score and the position of the box indicates the corresponding optimal node number. The optimal value is defined by the node number where the F1 score is close to its maximum value and the difference between precision and recall is close to the minimum value.

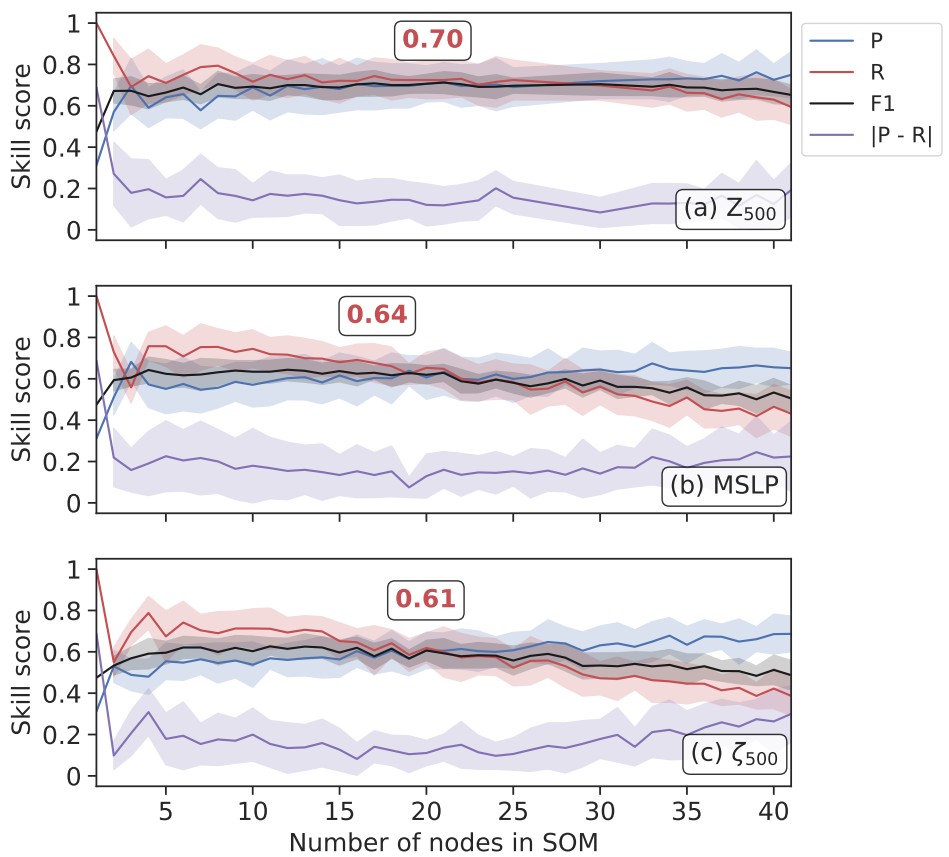

**Figure 8.** A comparison of the SOM blocking index performance for three variables in 101 years from the UKESM pre-industrial control period with a varying number of nodes in the SOM. Precision (P), recall (R) and F1 scores are calculated, and the absolute difference between precision and recall is also shown. Error bands show the standard deviation ($\pm 1\sigma$). The red number inset into each panel shows the optimal F1 score and the position of the box indicates the corresponding node number. As above, the largest F1 score is for $Z_{500}$, indicating that $Z_{500}$ is the best variable tested for analysing blocking patterns using the SOM-BI in UKESM.

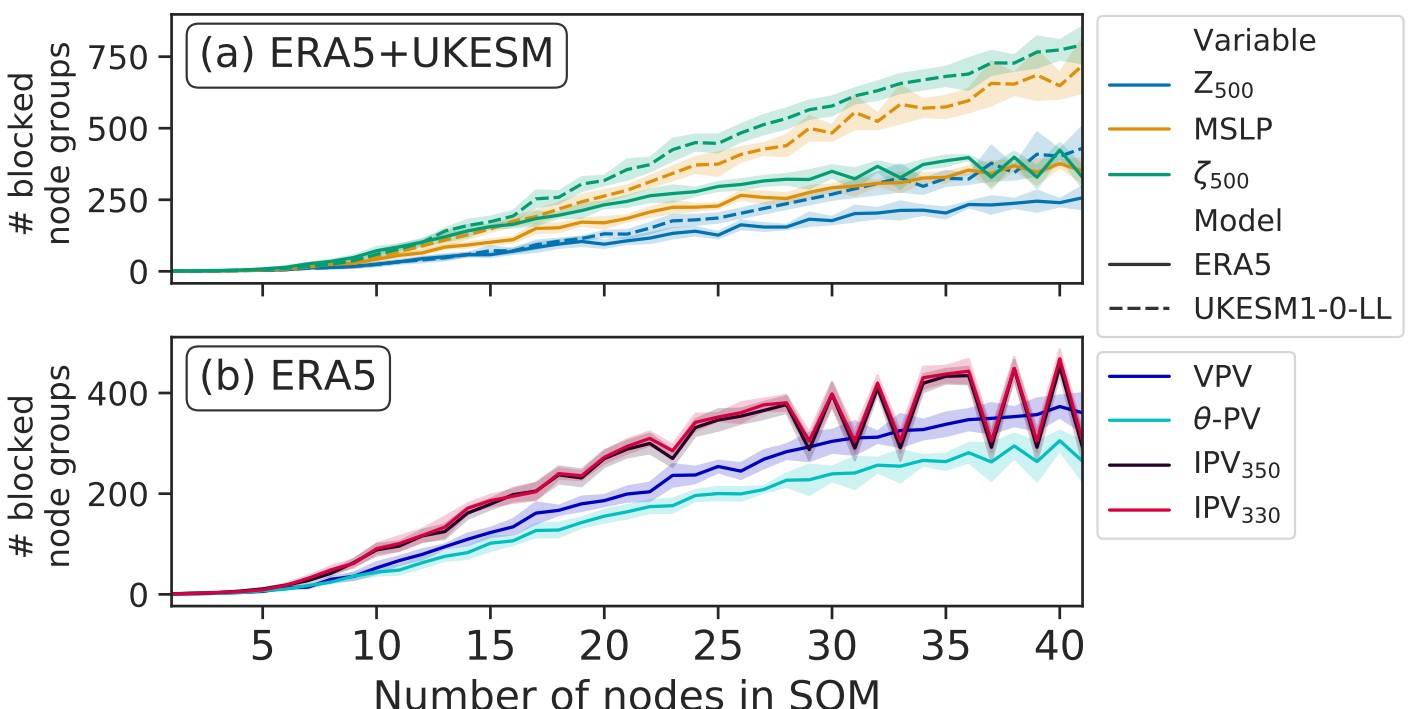

**Figure 9.** The number of node groups that are identified as blocked in the SOM-BI for ERA5 and UKESM for a range of node numbers and variables. The panels separate the variables available in both models from those only available in ERA5. Error bands show the standard deviation ($\pm 1\sigma$).

The results are shown for $1 \leqslant k \leqslant 41$. To measure out-of-sample skill, we used 10-fold cross-validation, where the GTD was

split into 10 separate sections for testing the SOM-BI. The SOM-BI is trained on nine of the ten data sections and its skill is evaluated on the remaining section. The skill scores shown only indicate how well the SOM-BI is able to predict the test period in question, which was not used for training. This ensures that the SOM-BI has not been tuned to the data we measure our skill against, which could give it an unfair advantage compared to the other BIs. For ERA5 we used 4 year periods (1980-1984, ... , 2015-2019 inclusive) to test on and trained on the remaining 37 years, with each 4-year period once serving as the independent

test set. In UKESM 10-year periods (1960-1959, ... , 2050-2059 inclusive) were used for testing the SOM-BI and it was trained on the remaining 91 year period. This 10-fold cross-validation procedure produces a range of precision, recall and F1 scores for each node number. Figures 7 and 8 show the mean values for precision, recall, F1 and the absolute difference between precision and recall. Figure 9 compares the number of groups of nodes identified as blocked for each variable. Error bands indicate the standard deviation of each skill metric ($\pm 1\sigma$).

Common features are observed for each variable for a very small or large number of SOM nodes. For small $k$ the SOM-BI identifies more days as blocked, such that R » P. This indicates that the SOM is under-fitting the data for European circulation

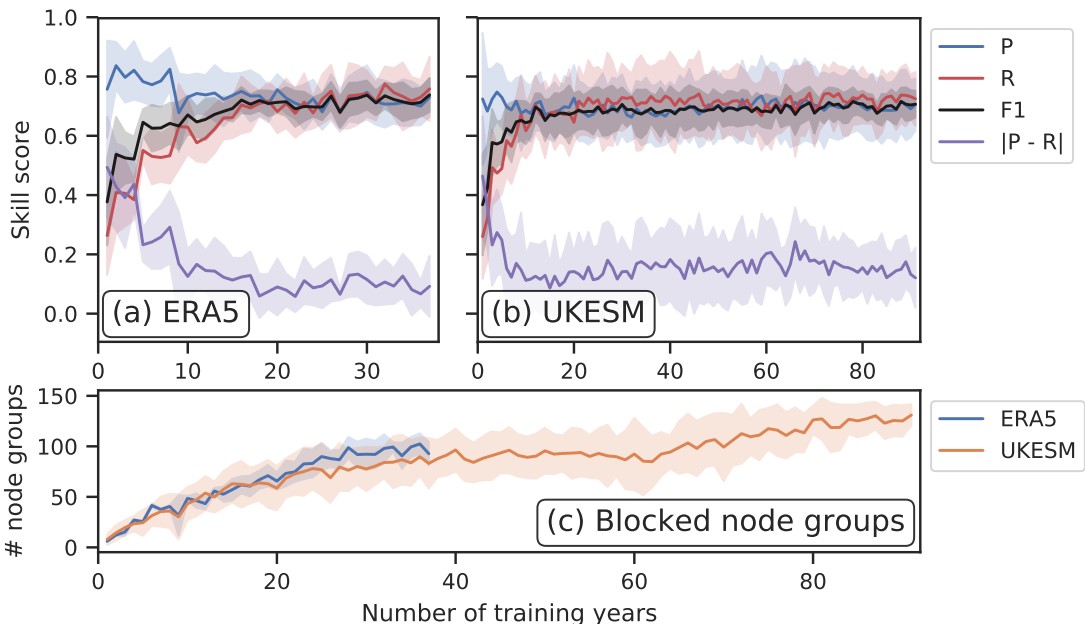

**Figure 10.** SOM-BI skill depending on the number of training years. (a) and (b) show the skill scores for ERA5 and UKESM, and (c) shows how the number of node groups associated with blocking varies with the length of the training record. 10-fold cross-validation is used, with 4 and 10 years used to test the SOM-BI for ERA5 and UKESM respectively. In both models $Z_{500}$ is the variable tested with 20 (5x4) nodes in the SOM. Error bands indicate standard deviation ($\pm 1\sigma$) in the skill scores depending on the training/test set combination.

patterns across the domain and so the algorithm lacks a precise delineation of blocking events. In other words, it could be beneficial to increase $k$ to be able to represent a larger number of dynamical states and thus to detect and describe blocking events more precisely. For large $k$, R « P, showing that the SOM-BI is trending towards overfitting the training data. We deduce

that the optimal $k$ occurs when the difference between P and R is small and the F1 score is close to its maximum value.

From Figs. 7 and 8 we find that for both ERA5 and UKESM the $Z_{500}$ anomaly is the best variable to use with the SOM-BI, with a mean F1 score of 0.74 and 0.71 for $k = 20$ and 21 in ERA5 and UKESM respectively. From Fig. 9a, $Z_{500}$ also shows the lowest number of blocked node groups for a given $k$, which shows that the blocked node groups are physically more explanatory in $Z_{500}$ than the blocked node groups associated with other for other variables, making the SOM-BI results easier

to interpret physically. MSLP is the second most effective variable, with an optimum F1 score of 0.66 and 0.64 for ERA5 and UKESM respectively. This lower peak performance is because the MSLP field has a lower signal-to-noise ratio as it is influenced by effects within the boundary layer such as heat lows. The PV-related variables exhibit a variety of lower skills, where the VPV field performing at a similar level to MSLP, since the vertical integration of the VPV variable enables it to capture the pattern of blocking better than other PV-based variables (Schwierz et al., 2004).

## 3.4 SOM-BI skill dependence on number of training years

One important verification for the SOM-BI is to ensure its robustness over long timescales. Contrary to the other BIs, the SOMs learn from training data. Therefore, the SOM-BI skill on test data will also be a function of how representative the training samples are of general states of the atmosphere. Here we investigate if the observational record, for example, is long enough to indeed ensure the same performance of the SOM-BI described above over longer timescales. For this purpose, we train the SOM-BI algorithm on a range of different numbers of training years, while keeping the number of years to test the algorithm performance consistent. Importantly, there is no overlap between the training and test data to ensure that the skill evaluation is truly independent, following the idea of statistical cross-validation (see e.g. Nowack et al., 2018; Mansfield et al., 2020). Figure 10 shows the results of this analysis for $Z_{500}$ and 20 nodes across both (a) ERA5 and (b) UKESM datasets, which is the best performing case according to our analysis above. Since the datasets have different lengths (41 years vs. 101 years), we tested the model on 4 and 10 years for each dataset respectively. For a small number of years, the algorithm only sees a few blocking events and so only identifies the particular node groups that are associated with these blocking events rather than identifying node groups that are in general associated with blocking events. This leads to a high precision for a small number of training years, particularly in the ERA5 data, since the SOM-BI is effectively over-fitting on a few events, but the recall and overall F1 score are low. This behaviour is confirmed by Fig. 10 (c), which shows that there is a small set of node groups associated with blocking for a small number of training years.

Figs. 10 (a) and (b) both show that the recall and F1 scores increase asymptotically for a larger number of training years, and the precision decreases asymptotically. These variations become very small after 20 years for both ERA5 and UKESM, which indicates that for around 20 years the SOM-BI seems to have achieved an optimal performance. Figure 9 (c) shows that the number of node groups associated with blocking continues to increase in both ERA5 and UKESM even after this point, with 120 node groups identified with blocking for UKESM over 91 years compared to 95 node groups over 37 years. However, these extra node groups occur rarely in the blocking datasets since they do not significantly affect either the precision or recall of the algorithm and are therefore not physically meaningful.

## 3.5 Cross-comparison of SOM-BI skill

For the SOM-BI to be effectively applied to understand future trends in atmospheric blocking, we need to verify that the training of the SOM-BI on the observational record is consistent with CMIP6 models. This step is necessary to ensure that the SOM-BI can identify blocking patterns in the models. If it is possible for the SOM-BI to identify blocking patterns in a CMIP6 model from training on the observations, then this shows potential for the SOM-BI to be applied consistently across a model ensemble. Furthermore, if the SOM-BI can be trained on a CMIP6 model and tested on the observations, differences in the skill of the SOM-BI would highlight limitations in that model's ability to represent blocking patterns. This could be applied across a model ensemble to compare the skill of different models at representing blocking patterns.

To investigate the feasibility of such studies, we test the skill of the SOM-BI algorithm by training $Z_{500}$ data on the 41 years from the ERA5 dataset and testing on the UKESM and vice versa. Table 2 shows the differences in the optimal performance

| Training dataset | Test dataset | F1 score | Number of nodes | Number of blocked node groups |
|:---:|:---:|:---:|:---:|:---:|
| ERA5 | ERA5 | 0.74 | 20 | 95 |
| UKESM | ERA5 | 0.74 | 21 | 134 |
| UKESM | UKESM | 0.71 | 20 | 131 |
| ERA5 | UKESM | 0.71 | 19 | 99 |

**Table 2.** A comparison of the optimal F1 score for when $Z_{500}$ ERA5 and UKESM datasets are trained and tested on themselves and each other respectively. The corresponding node number and number of blocked node groups is shown. When the dataset is tested on itself, 10-fold cross-validation is used and the mean value is shown. The optimal F1 score is identified by finding the node number with the smallest difference between precision and recall whilst maintaining a relatively high F1 score.

for $Z_{500}$ across the different datasets. In all cases several node numbers were tested, and we identified an optimal node number of 20 or 21 across all the configurations of training and testing data. There was also a good performance of the SOM-BI for
other node numbers that is consistent with Fig. 7a (not shown). The stable performance of the SOM-BI shows that there is a consistent range of synoptic weather patterns between the ERA5 and UKESM for European summer. It also indicates a consistency between the labeling that occurred in the GTD across ERA5 and UKESM, despite the reduced performance in the blocking indices to label the GTD. This further shows that UKESM describes blocking patterns in a similar enough way to the historical observations for useful study of blocking events, which in turn reinforces the validity of studies in blocking trends
from the CMIP6 archive (Davini and D'Andrea, 2020). Finally, this underscores the potential for the SOM-BI to be used in understanding future trends and diagnosing model skill across the CMIP model ensemble.

### 3.6 Dependence of SOM-BI skill on other parameters

Apart from the SOM node number, number of years trained over and training dataset, there are several other parameters that could be modulated within the SOM-BI framework. First, we investigated 5-fold cross-validation on the ERA5 dataset, which
involves testing the SOM-BI on 8 years of data five times. This was found to have a marginally lower performance than 10-fold cross-validation. Furthermore, we tested an alternative approach to identifying the corresponding best matching unit for the SOM pattern, where we only used positive anomalies to define the BMU. Since we are only interested in positive anomalies it is possible that such an approach would increase the skill score, particularly for events such as the 2019 European heat wave (Fig. 6). However, this modification was found to have a negligible effect on the overall SOM-BI skill.

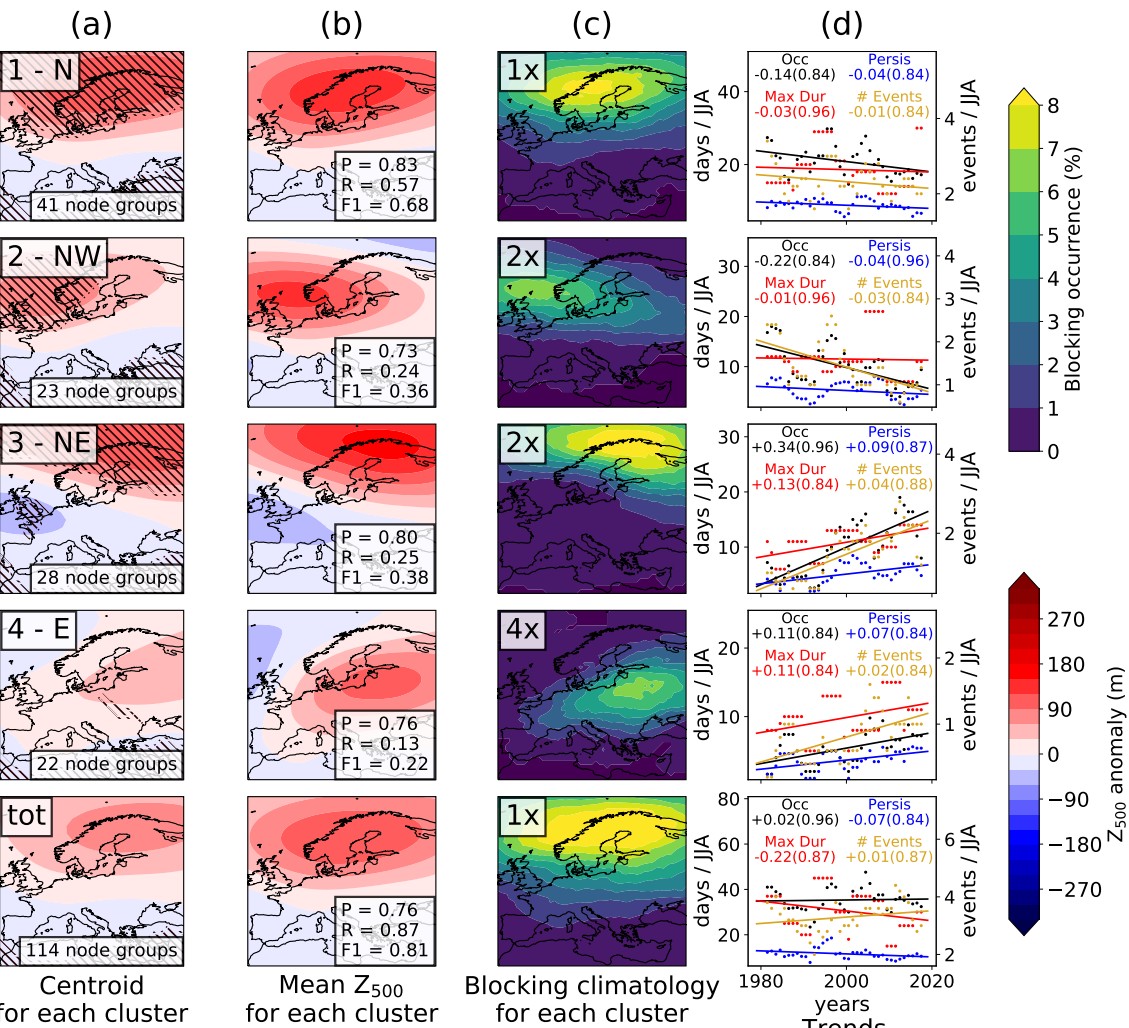

**Figure 11.** Application of the SOM-BI to ERA5 $Z_{500}$ 1979-2019 JJA, using the method outlined in section 2.7. (a) The K-means centroids of the mean codebooks. Hatching indicates regions where the mean codebooks for contributing node groups agree on the sign. The number of node groups associated with each cluster is indicated in the bottom right of each panel. The labels in the top right of each panel reflect the main region with a positive anomaly. "tot" is the total combination across all blocked node groups. To show that the four clusters are consistent with the fields they represent, we show in (b) the average $Z_{500}$ field across all days belonging to each cluster of blocked node groups. At the bottom right of each panel in (b) the precision, recall and F1 scores are shown for each cluster. (c) The blocking climatology for each set of node groups, derived using the DG83 index for each blocked day identified by SOM-BI for the given K-means cluster. Since the frequency of blocked events varies strongly between each cluster, the climatologies have been scaled by the numbers in the top left of each panel. (d) Historical trends as characterized by four different metrics for each cluster, using five-year moving average data: black - occurrence of pattern; blue - persistence (average duration) of pattern; red - maximum duration of block; gold - average number of blocked events (uses the right hand y-axis scale). The numbers (numbers in brackets) show the gradients (p-values) for each trend, which are all insignificant after correcting for multiple testing and autocorrelation.

## 3.7 Application of the SOM-BI to ERA5

A central question of current research is how the characteristics of regional blocking events are affected by climate change (Woollings et al., 2018; Drouard and Woollings, 2018; Kornhuber et al., 2019, 2020). Here, we briefly demonstrate how the SOM-BI can be used to study such effects. For this purpose, we apply SOM-BI to ERA5 data (Fig. 9a), where our optimization yields the best performance for 20 SOM nodes (using $Z_{500}$), and a total set of 114 blocked nodes (section 7). Clearly, this large number of different blocking patterns with subtly different characteristics creates a challenge for easy interpretation of the results. In addition, because several of these node groups occur very infrequently, perhaps only once in the 1979-2019 period, a meaningful study of their trends is not always possible. We therefore need to develop a methodology to sensibly match node groups to more general types of regional blocking patterns, which we can then study instead. We here suggest a post-processing approach using the aforementioned K-means clustering analysis (Diday and Simon, 1980), but now applied to the 114 mean codebooks (section 2.7) identified for each node group. The goal of this post-processing step is to identify distinct 'K-means clusters of SOM node groups, where each K-means cluster summarizes a pattern-wise similar set of blocked SOM node groups. In Fig. 9 (a), $k = 4$ is chosen since it is a common choice for identifying weather regimes (Michelangeli et al., 1995; Cassou, 2008; Ullmann et al., 2014; Strommen et al., 2019; Fabiano et al., 2021), but we note that a larger value of $k$ yields a more detailed classification of blocking patterns. Hierarchical clustering was also tested as an alternative to K-means, and similar patterns were produced (not shown).

To illustrate the method, Figure 11 (a) shows the cluster centroids for each set of mean codebooks for this case of $k = 4$. The bottom panel shows the mean pattern across all centroids. Regions of hatching show where all of the mean codebooks in each centroid agree on the sign. The number of node groups included in each cluster has been labelled in the bottom right of Fig. 11 (a). The different clusters show distinct regions of positive anomalies, with generally strong agreement across all mean codebooks included in each cluster. This underlines that the clustering approach is effective at identifying distinct types of blocking events. The clusters have been labelled in the top right of Fig. 11 (a) reflecting in each case the main region of positive anomaly. Fig. 11 (b) shows the mean $Z_{500}$ field across all blocked days identified for each cluster, which are highly consistent with Fig. 11 (a). Consequently, the subsets of node groups are as expected physically consistent with the circulation patterns across these blocked days so that the K-means clusters can indeed be used to study specific types of regional blocking. In Fig. 11 (c), we show the blocking climatology associated with each cluster derived using the DG83 index across the days identified as blocked in each cluster. The climatology again matches well the patterns identified in (a) and (b). The bottom panel of Fig. 11 (c) shows the total blocking climatology from the SOM-BI. It is similar to the corresponding DG83 blocking climatology in Fig. A4 (b), with a slightly reduced number of blocking events around the Scandinavian high. This suggests that the SOM-BI is reducing the identification of events on the edge of the domain.

Having established that SOM-BI, following by K-means post-processing, identifies clusters representing distinct regional blocking events, we turn our attention to studying potential trends in such blocking patterns. Such trends are given in Fig. 11 (d), using four different metrics to characterize the events. These are the rate of occurrence of events ("Occ"; the number of days in JJA that are associated with the pattern each year) and their persistence ("Persis"; the average persistence of a blocking

pattern in days), maximum duration ("Max Dur"; the longest event for each year), and the number of events ("# Events"; the total number of continuous blocked events per year). These quantities are calculated for each cluster and we average all metrics using a five-year centred moving window, which is necessary to ensure that at least one event for each of the four blocking patterns occurs within a given period.

The numbers (numbers in brackets) show the gradients (p-values) for each trend. The p-values have been corrected for autocorrelation using the Zwiers and von Storch (1995) two-tail Student's t-test, and the multiple hypothesis testing has been corrected for using the false discovery rate (Benjamini and Hochberg, 1995; Horton et al., 2015). Since all of the p-values are large, none of the trends are significant. However, we note that the number of E and NE European blocking events doubles whilst the number of W European events halves across the 1979-2019 period. Whilst none of these trends are statistically significant, it highlights how the SOM-BI method could be used to study changes in the characteristics of European blocking over time, e.g. in longer climate change simulations. The SOM-BI can provide information that is not directly available in the other BIs discussed here, since it can be used to study long-term trends across several types of automatically identified European blocking patterns.

## 4 Discussion and Conclusions

Using self-organizing maps (SOMs), we have developed a new blocking index (SOM-BI). This has involved the creation of a new time series dataset (GTD) to describe when blocking events have historically occurred over a region. We have described our approach as unsupervised through its use of the SOM algorithm, but we note that the SOM-BI also employs supervised learning through its blocking classification using the GTD. By studying the case of European summer, we have identified a similar or better skill score for SOM-BI compared to several other blocking indices (BIs) using ERA5 reanalysis data from 1979-2019. We further applied our new approach to a pre-industrial control run from UKESM, and find that our method shows consistent skill for this model dataset, whereas the other BIs substantially lose performance in this case. Whilst no individual SOM node directly corresponds to a weather regime such as blocking, with an optimal node number we can develop a set of node groups which are associated with blocking. We have also found that 20 years are needed to train the SOM-BI, which underlines that the SOM-BI has a robust level of performance if trained on standard reanalyses or on typical lengths of climate model simulations. The performance of the SOM-BI is also robust to the dataset used to train it, since it shows good performance when it is trained on the ERA5 data and tested on UKESM and vice versa. These results show that unsupervised learning can be usefully applied to understand regional blocking events, both historically and in the future.

We have further compared the performance of SOM-BI for a range of variables in both ERA5 and UKESM that have been classically used to study blocking (Figs. 7 and 8). We find that the best skill is obtained when applying SOM-BI to the $Z_{500}$ field because it exhibits the best signal-to-noise ratio in blocking identification. This is reflected in Fig. 9, which shows that for a given node number the $Z_{500}$ SOM-BI identifies blocking patterns with a smaller number of node groups than for other variables.

We have confirmed that individual SOM nodes do not represent weather patterns perfectly so that care needs to be taken in using SOM patterns as a means of diagnosing weather patterns (Gibson et al., 2017b). If individual SOM nodes were used to create a blocking index, or if a small node number was used (3-6 nodes) there would be a high recall and low precision in detecting blocking using this approach, which would be the equivalent to some of the approaches applied elsewhere (e.g.
Horton et al., 2015). If a higher node number (12+) was used and only one node was associated with blocking, then there would be a high precision and low recall, and overall a lower F1 score than for a low node number. However, by using a large number of nodes and studying groups of nodes across periods of five days, we have developed an algorithm that can regionally identify blocking patterns with optimal precision and recall, and which outperforms several conventional blocking indices for this task.

Using this algorithm has involved the creation of a GTD, a binary dataset that identifies regional blocking events. There
are several limitations to this approach. Firstly, the choice of domain is somewhat arbitrary and here primarily motivated by a specific scientific question (summer heat waves in Europe), and events which are on the edges of the domain are excluded, even though a large region within the domain could be considered blocked. In addition, the task of assigning a binary label to each day can be further complicated, since there is subjectivity in assigning a binary label to the onset and decay of blocking events. However, by focusing on events which are centered within the domain, a broad agreement with the identification of blocked
events was achieved, despite the subjective nature of this approach. The fact that the SOM-BI exhibits consistently good skill across ERA5 and UKESM even when the SOM is trained on the other dataset underscores the validity of our labelling applied to both model and reanalysis data.

The use of SOMs as a blocking index provides opportunities for regional study that are not directly available in the other BIs. Through an additional post-processing step involving K-means clustering on blocked node groups (sections 2.7 and 3.7),
we have shown that the SOM-BI can identify specific types of blocking events and provide detailed information about the changing nature of blocked events over a European subdomain (Fig. 11). The case of $k = 4$ has been shown in Fig. 11, but larger values of $k$ can also be chosen to identify more distinct types of blocking pattern. Whilst the SOM-BI does not directly produce a gridded climatology of blocking patterns, we have shown that the SOM-BI can be integrated with the other BIs to develop a climatology that only considers only those days detected as blocking by the SOM-BI. This results in a SOM-BI
climatology with a higher precision than the BI climatology.

We intend to apply this method to future trends across CMIP5 and CMIP6 models to better understand the patterns of blocking in models, diagnose model skill at reproducing the historic patterns of European circulation regimes and compare projections of future changes in blocking patterns. The identification of distinct blocking patterns from node groups enables a detailed study of blocking characteristics over European subdomains as shown in Fig. 11. Further quantities such as the
Rossby wave breaking properties or the nature of blocking onset and decay can also be studied. This could be done by studying particular dynamical quantities on the blocked days identified by the SOM-BI, and extended by contrasting the dynamical quantities across different categories of blocking pattern identified by the SOM-BI node groups. We also make our GTDs available for both ERA5 and UKESM, which have wider application in understanding historic blocking events, how they interact with other meteorological phenomena (such as heat waves and droughts) and comparing blocking patterns between
reanalyses and CMIP6 models (cf. process-based climate model evaluation, Nowack et al. (2020)). We consequently encourage

similar ground truth datasets to be created for other world regions and seasons, and highlight that our method could then be trained for and applied to those regions.

*Code and data availability.* The scripts used for the self-organizing map blocking index, the plots for this paper and the ground truth datasets for labelling of blocking events in JJA Europe (in both ERA5 1979-2019 and UKESM pre-industrial control 1960-2060) can be accessed in 595 github.com/carlmagnusthomas/SOM-BI. ERA5 data is available from confluence.ecmwf.int and UKESM data is available from esgf-node. llnl.gov

## Appendix A: UKESM case studies

In the UKESM pre-industrial run, we show in Fig. A2 part of a heat wave in the (arbitrary) year 2014. This year shows the largest spatial extent of heat extremes, where the number of grid cells exceeding the 90th (99th) temperature percentile peaks at 600 66% (24%) on 19 (20) July 2014. To complement this extreme case, we also show in Fig. A3 a period from the 2030 summer, which shows the edge of a blocking pattern in Eastern Europe on the 19 July and an anticyclone shifting across Europe over 20-27 July.

Since VPV is not available as a variable, the S04 blocking index cannot be calculated, and we have instead shown MSLP in Figs. A2b and A3b.

Many of the same features are observed. Extreme heat is associated with persistent high pressure and stationary surface winds. The MSLP field is broadly correlated with the $Z_{500}$ anomalies, but frequently the $Z_{500}$ anomaly doesn't represent the MSLP anomalies well, such as on the 25th July 2014 shown in Fig. A2, where low MSLP is contrasted with high $Z_{500}$. The AGP index in general performs worse than in ERA5, since the zonal $Z_{500}$ gradients are not as prominent. The DG83 index is still able to describe blocking patterns from the $Z_{500}$ anomalies. The SOM-BI labelling is generally consistent with the ground 610 truth dataset in both cases.

The MSLP field is broadly correlated with the $Z_{500}$ anomalies, but frequently the $Z_{500}$ anomaly does not represent the MSLP anomalies well, such as on the 25th July 2014 shown in Fig. A2, where low MSLP is contrasted with high $Z_{500}$. The surface wind fields in UKESM similarly show the easterly wind direction associated with high pressure and vice versa, particularly when the MSLP anomalies are also strong such as on the 20-21 July 2030 in Fig. A3.

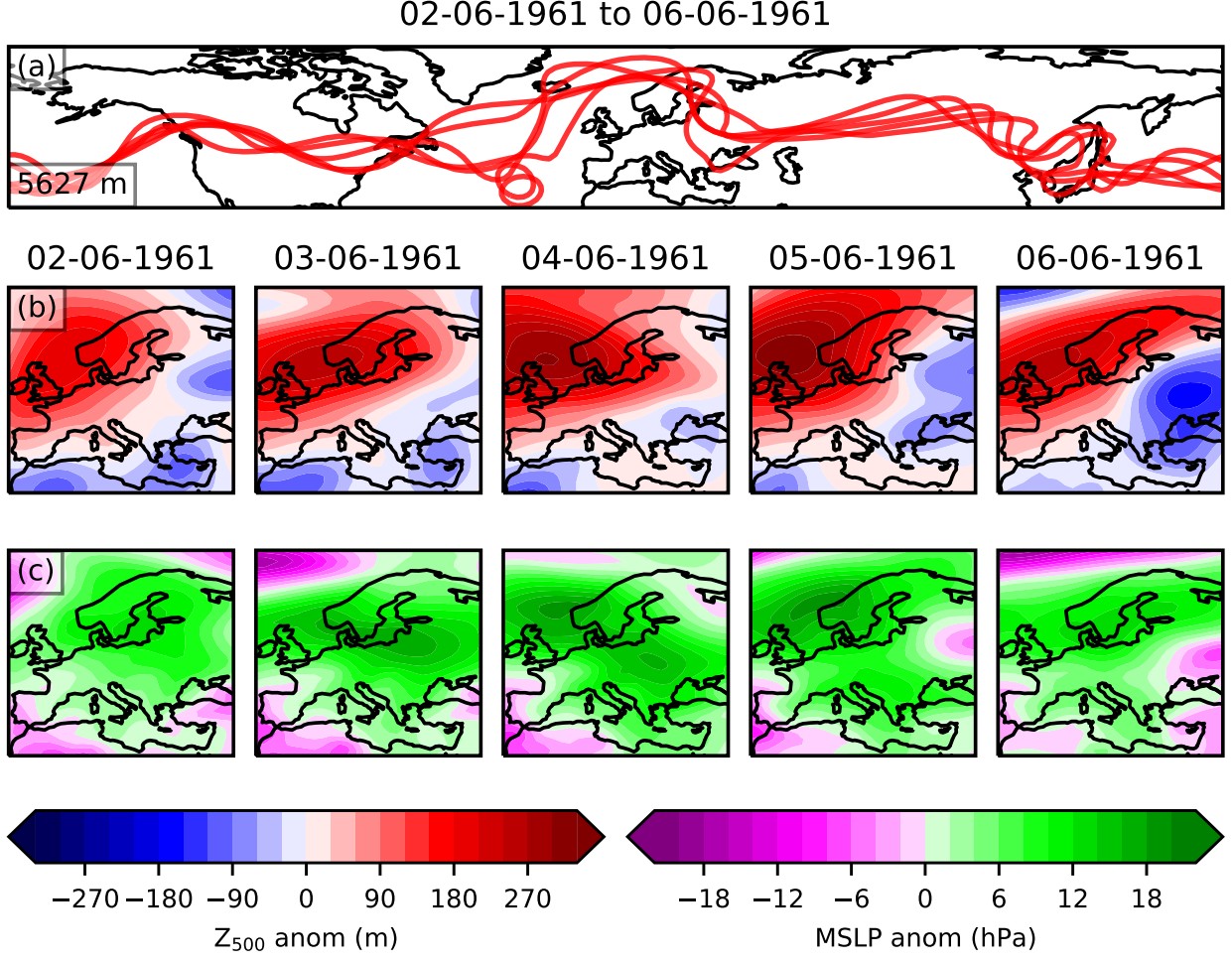

**Figure A1.** The information used to classify blocks in the UKESM GTD. (a) shows the daily $Z_{500}$ contour for the averaged value across 30-70 °N, indicated in the bottom left of the panel. (b) and (c) show the $Z_{500}$ time detrended anomaly and MSLP anomaly for each day.

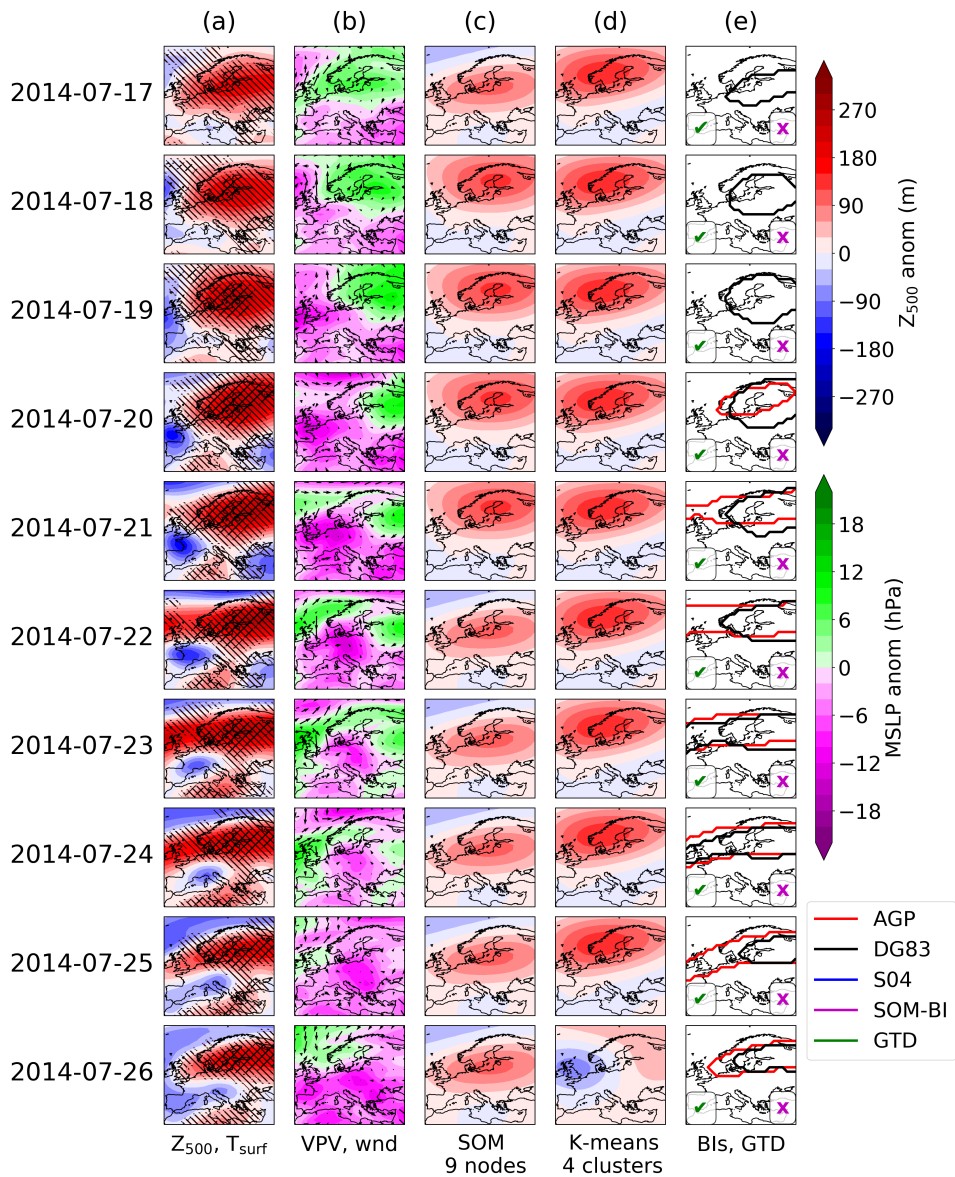

**Figure A2.** As with the case studies shown in figures 5 and 6, but for a heat wave in UKESM.

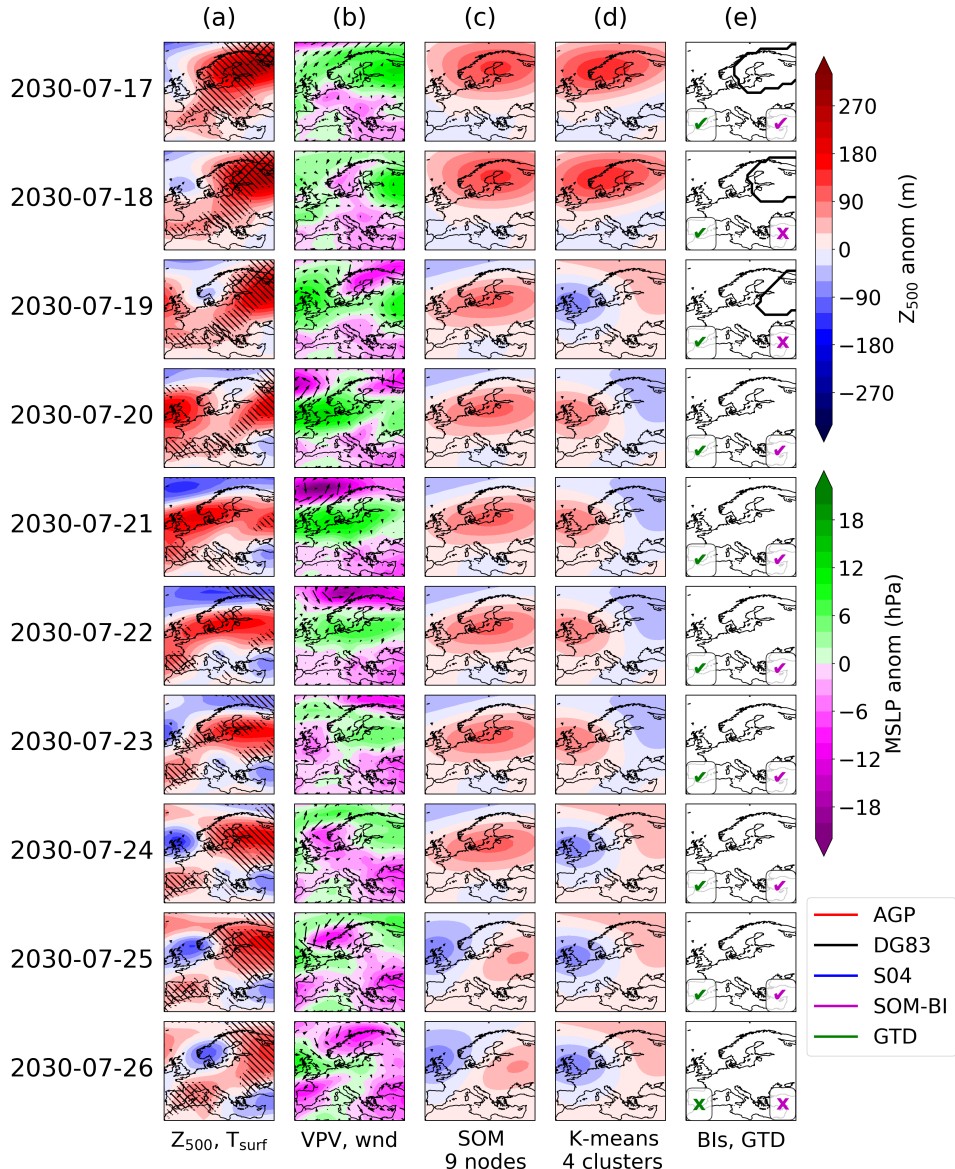

**Figure A3.** As above, but for a transient period in UKESM.

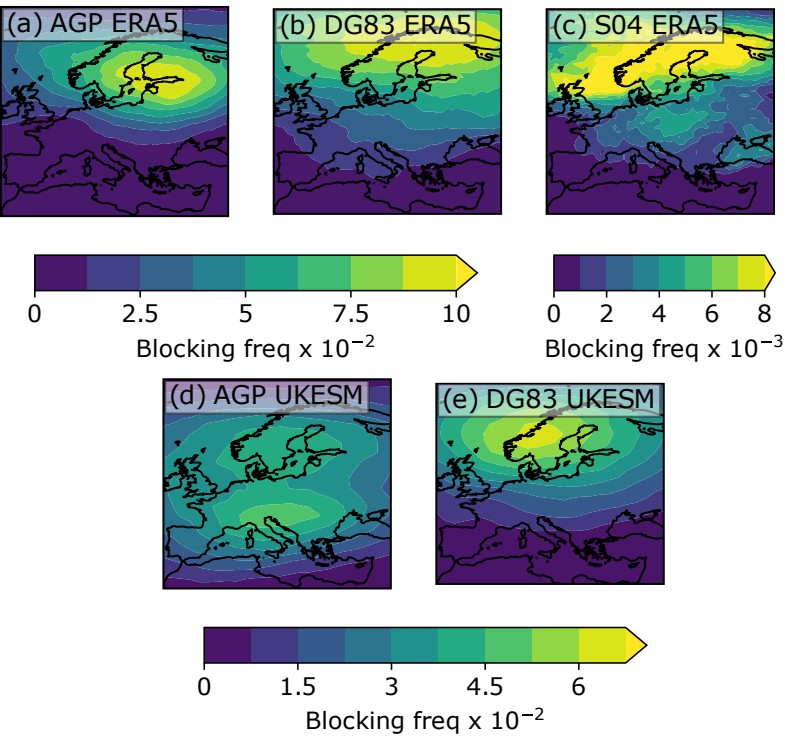

**Figure A4.** Occurrence of blocking events per grid cell across JJA Europe for three BIs in ERA5 1979-2019 and two BIs in UKESM JJA 1960-2060.

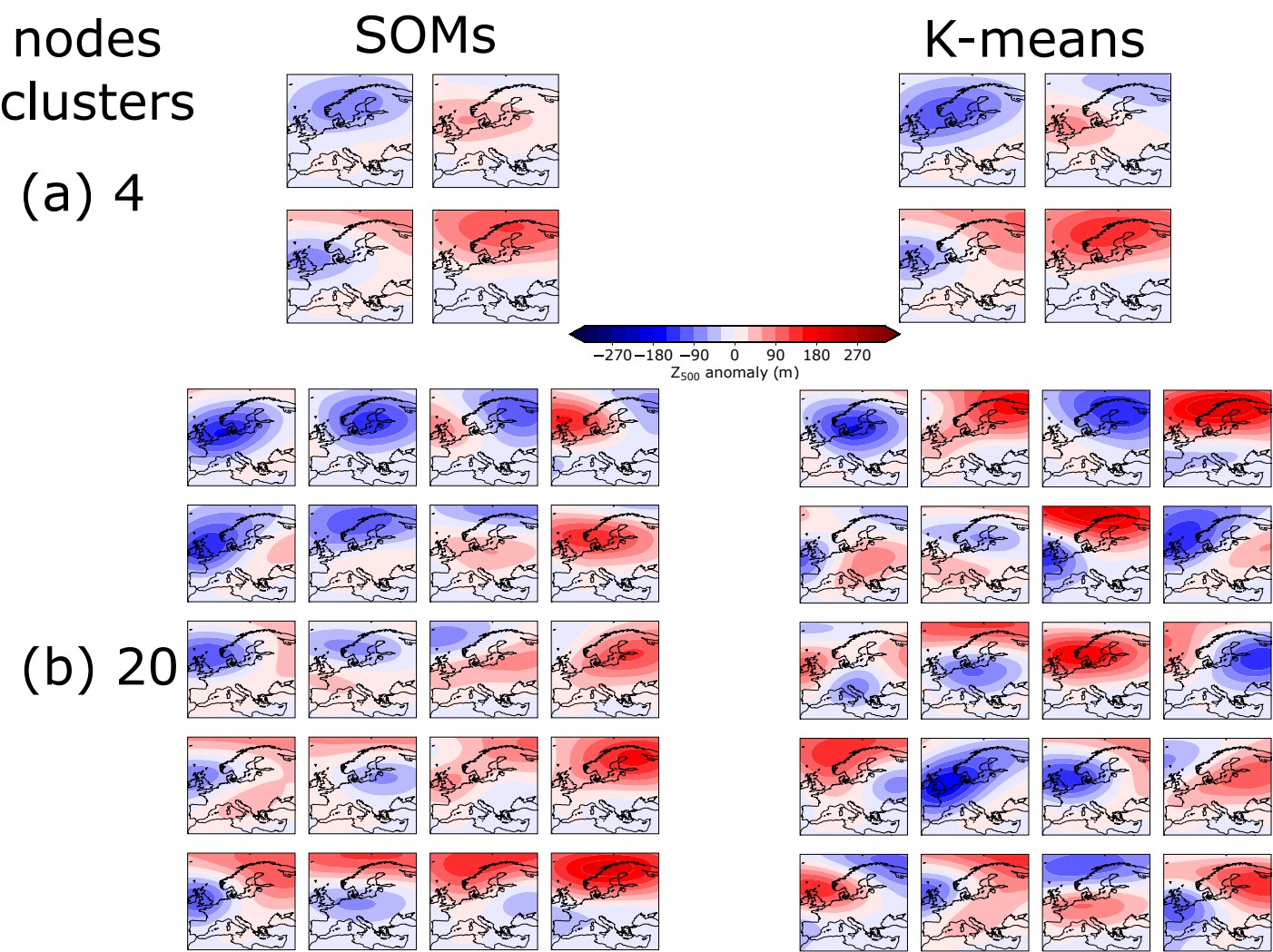

**Figure A5.** Comparison of the SOM analysis and K-means clustering for (a) 4 and (b) 20 nodes/clusters. Whilst the K-means and SOM analysis produce a similar set of patterns for smaller node numbers, their behaviour diverges for larger node numbers. Since the neighbourhood function ensures that several nodes are updated at once, the SOM produces a continuum of weather patterns. However, since the K-means clustering updates each centroid independently, it will seek to maximise the differences between each cluster. This causes some centroids for high cluster numbers to represent mixed weather regimes that are less realistic than the SOM continuum of weather regimes. The data used for above is ERA5 $Z_{500}$ across JJA 1979-2019.

*Author contributions.* CT developed the SOM-BI method and labelled the GTDs under the supervision of PN and AV. GL developed the code for the SOMs and JH provided input into the discussion of the use of SOMs. CT wrote the initial draft of the paper and all authors contributed to its review and refinement. PN suggested the study.

*Competing interests.*   No competing interests are present.

*Acknowledgements.*   PN is supported through an Imperial College Research Fellowship. AV is partially funded by the Leverhulme Centre for Wildfires, Environment and Society through the Leverhulme Trust, grant RC-2018-023. CT is funded by the Natural Environment Research Council SSCP DTP, grant NE/L002515/1. The authors would like to thank two anonymous reviewers who have contributed significantly to improving the quality of the paper.

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
