# Peer review of "An unsupervised learning approach to identifying blocking events: the case of European summer"

_Weather and Climate Dynamics, 2021_

## Author Comment (AC2)

[revised manuscript text omitted]

This paper focuses on the development of a new blocking index that is based on self-organizing maps (SOMs) and a ground truth dataset (GTD). We call this the self-organizing map blocking index (SOM-BI). Different aspects of the development of this SOM-BI are summarised in the following subsections:

95
- **Section 2.2**: using geopotential height and potential vorticity or mean sea level pressure fields we develop a ground truth dataset that defines blocking patterns across the ERA5 reanalysis and a UKESM pre-industrial control run.

- **Section 2.4**: we apply SOM analysis to identify a representative set of circulation patterns from a pre-defined number of nodes. Each day of data is assigned to its best-matching node.

- **Section 2.5**: we compare the nodes with the GTD to identify node groups which are associated with blocking patterns
100 across five-day periods. The SOM-BI identifies a set of node groups which are associated with blocking. The skill scores used to identify these node groups are described in **section 2.6**.

- **Section 2.7**: from the total set of blocked node groups identified in the SOM-BI, we categorise subsets of blocked node groups using K-means clustering. This enables the SOM-BI to identify different types of blocking events within the domain and to study trends in their occurrence.

105 The data we use is described in **section 2.1**. We compare the SOM-BI to other BIs and K-means in **section 2.3**. This comparison is carried out for case studies (**section 3.1**) and blocking identification skill (**section 3.2**). **Sections 3.3 - 3.6** describe the optimization and validation of the SOM-BI across a variety of hyperparameters and input datasets. An example application of the SOM-BI to the ERA5 reanalysis to study trends in the occurrence of specific types of blocking events is given in **section 3.7**.

[revised manuscript text omitted]

---

## Author Response (AR2)

**Response to Reviewer #1 minor revisions Draft 1**

*The authors greatly improved the manuscript addressing the most of my comments with complete and accurate replies. I have just a few minor concerns that I list here below:*

We thank the reviewer for the positive review and the helpful comments.

*L23: You said that your method is an unsupervised learning: although this is true for SOM, however you exploit a ground truth to define which node found by SOM is blocked or not. Would be more correct to call it as a supervised learning?*

While our method uses the ground truth dataset to identify SOM node groups associated with blocking (as we indeed need a label for blocking eventually), we think that it is helpful to highlight the contrast between our method and direct regression approaches, which would simply learn the relationship between predictors and blocking in a supervised sense without prior unsupervised clustering step.

We have included a sentence in L540-542 to expand this definition:

"We have described our approach as unsupervised through its use of the SOM algorithm, but we note that the SOM-BI also employs supervised learning through its blocking classification using the GTD."

*L63: Even if I found it fun to read, I wonder if the pronunciation is really necessary…*

While we agree that this pronunciation is not strictly necessary, we realise that our manuscript is of rather technical nature. Consequently, our goal here is to create a memorable name, which helps readers remember our new method.

*L78: what do you mean with skill here? Please rephrase.*

Skill here refers to blocking identification skill, F1 score in particular, which is defined in section 2.6. However, the word is not needed here so that we simply removed it to avoid confusion. The sentence in L77-78 now reads:

"We identify a moderate improvement in blocking identification over the BIs for the reanalysis period and a significant improvement for the UKESM data."

*L91-109: the paper is already quite long, the outline of the method section could be removed.*

Thank you. We agree that the outline of the method section has been added to help the reader navigate the lengthy manuscript.

Thank you. We have removed the outline of the methods section.

*L121: relative vorticity?*

Thank you for pointing out. "Relative" has been added to L103.

*L164: I would remove Figure 2. It is true that you analyzed MSLP instead of PV, but the approach to define the GTD is the same.*

It is true that Figure 2 shows the same process of labelling with a minor difference of variable, but it also shows the labelling procedure in a climate modelling context so we still think that it should be included in the manuscript. We have therefore moved the figure into supplementary information as Figure A1.

*L215: Why fifth? I count three blocking indices, shouldn't this be the fourth?*

Thank you. L202 has been amended accordingly.

*Figure 3: Please write the BMU acronym in the caption. The 10^-3 multiplication is unclear: do you mean that the initial SOM anomaly has a +-270000 meter anomaly?*

We have added a definition in the caption. The initialised pattern in step 1 does indeed have very large anomalies, and the caption of Figure 3 includes a clarifying statement. The caption has changed to:

"The self-organizing map algorithm. Shown using a 3x3 node SOM with ERA5 Z500 JJA 1979-2019. The PCA-initialised SOM pattern (step 1) has a much larger amplitude so has been multiplied by 10^-3 for visualisation purposes. The BMU refers to the best-matching unit, the SOM node which most closely matches the sample day."

*L265: here and in other instances you refer to "5-day period". However blocking events in the GTD can be longer, is this correct? Therefore why not calling them blocking events or something like that? More in general, although some clarification in the new version has been presented I still find confusing the methodology for the blocking definition in the GTD. For example, at L268 you say: "Since each five-day period has been classified either as blocked or not blocked, [...]". This sentence makes no sense to me, since EACH DAY should be defined as blocked or non-blocked (even if this definition is based on a 5-day threshold). It seems to me that here you are trying to identify which node describes each blocking event, but blocking events last more than five days.*

It is true that blocking events frequently last more than five days. This is reflected in our GTD, which labels each day as blocked or non-blocked. However, since a blocking event cannot last for less than five days (a common persistence criteria) we need to study groups of five days to identify for each individual day if a day is blocked. One cannot identify a block by studying a map for a single day in isolation.

By studying every consecutive five-day period across the dataset, we can see if a block persists across that period or not. This will identify every blocking event across the dataset. Blocking events longer than five days are identified in this way from the identification of overlapping five-day periods as blocked.

For example, if two consecutive five-day periods are labelled as blocked (across days 1-5 and days 2-6), then there is a blocking event that lasts six days (across days 1-6 inclusive). If fifteen consecutive five-day periods are labelled as blocked, then a blocking event of 20 days in duration has been identified.

Therefore, by studying every five-day period we identify all blocking patterns across the ground truth dataset.

We have amended the explanation by adding a sentence in L145-147:

"Blocking events longer than five days are also identified through this approach, since days that are part of any consecutive five-day blocked period are labelled as blocked. Blocking events longer than five days are then identified through a series of adjacent five-day blocked periods."

***L381: I still feel this approach is overly inclusive for blocking events. The climatology you get from Table1 is quite unrealistic, with AGP having 60% of blocked days, twice than DG83 – which is usually detecting much more areas as blocked (see Fig 6-7) and have larger climatological frequencies than AGP (Fig 2 in Woollings et al 2018). Usually spatial and temporal criteria are applied to blocking indices to avoid detection of spurious events, and this does not seem to have been done here.***

Thank you for spotting this error. We have corrected the calculation of the AGP index so that we have now obtained a blocking occurrence of 20%. The numbers in the table, relevant figures (Figs 5, 6 and A4) and references in the text (L348-349 and L383) have been updated. The spatial and temporal criteria have been applied (see the response to your final comment).

***L382: please reword: "if a blocking event has been identified" -> "if a single grid point has been identified as blocked".***

This is a misunderstanding. We do not identify blocking events across Europe on the basis of a single grid point, but if a blocking event has been identified within the European sector. This requires the block to meet all of the various thresholds for persistence, area and overlap. L369-371 says "if a blocking event has been identified within the European sector. A blocking event is not identified if the thresholds for amplitude, persistence, area and overlap discussed in section 2.3 are not met within the European domain." – these statements accurately reflect our analysis.

***L383: which threshold in amplitude, persistence area and overlap are you referring to? In Section 2.3 only the indices are defined, no discussion of the tracking is introduced.***

In L 170-192 in section 2.3 the amplitude, persistence, area and overlap thresholds are referenced. We have expanded this section to refer to the specific criteria, and to limit the length of this manuscript we refer the reader to Pinheiro et al. (2019) for an extensive discussion of these criteria:

"All of these methods have been further developed by Pinheiro et al. (2019) who applied four thresholds for each blocking index: the magnitude of the anomaly, the persistence of the blocking event (minimum five days), a minimum area over which the anomaly takes place (10^6 km^2) and an overlap criterion which measures if there is continuity across the blocked region between different days (an overlap of the blocked contours). We adopt their thresholds and as such study the three indices compared in Pinheiro et al. (2019) including their modifications:"

> – **AGP** - the geopotential height gradient method, which is the Tibaldi and Molteni (1990) index as adapted by Scherrer et al. (2006) to construct a two-dimensional field of geopotential height gradients

– **DG83** - the Dole and Gordon (1983) method of investigating positive geopotential height anomalies

– **S04** - the Schwierz et al. (2004) method of identifying persistent anomalies in the potential vorticity field (VPV) averaged over 150-500 hPa (VPV).

We refer the reader to section 2.2 in Pinheiro et al. (2019) for a detailed discussion of these methods and their associated thresholds. However, our analysis differs from the methodology…"